# Critical length screening enables 19% efficiency in thick-film organic solar cells

Yuan Meng[1], Bo Cheng[2], Dongcheng Jiang[1], Jiangkai Sun[1], Jiawei Qiao[1], Beibei Shi[1], Haisheng Ma[3], Jingtian Zhu[1], Lianbo Wang[1], Runzheng Gu[1], Peng Lu[1], Yanna Sun[4] ✉, Xiaoyan Du [1], Xia Guo[2], Ke Gao[4], He Yan [5], Maojie Zhang [2] ✉, Feng Chen [1], Yanming Sun [3], Xiaotao Hao [1] ✉ & Hang Yin [1] ✉

The commercialization of organic solar cells (OSCs) requires thick-film active layers, yet current thick-film-compatible acceptor selection based on zero-field mobility is unreliable due to methodological inconsistencies in experimental protocols, fitting models, and single-carrier device configurations. Existing literature indicates that the zero-field mobility in high-performance thick-film devices shows negligible differences compared to thin-film counterparts, thereby invalidating its significance as a selection criterion. This study introduces a protocol identifying critical length - an intrinsic property distinct from zero-field mobility - as the decisive factor for thick-film OSC performance. Comparative studies reveal that enlarged acceptor domains with high critical length yield increased hopping frequency, improved charge mobility and reduced field-dependent, collectively enhancing performance. Applying this criterion, we identify BTP-eC9 as a general acceptor, achieving 19.0% efficiency in thick-film D18:L8-BO:BTP-eC9 OSCs. This work not only demonstrates the fabrication of high-performance thick-film OSCs, but fundamentally advances material screening methodology specifically tailored for thick-film-compatible organic semiconductors.

The development of high-performance thick-film organic solar cells (OSCs) is critical for scalable organic photovoltaic technologies[1–3]. Active layers exceeding 300 nm thickness enhance absorbance by improving photon harvesting specifically in the long-wavelength region, thereby directly increasing short-circuit current density ($J_{SC}$). Furthermore, such architectures reduce sensitivity to nanoscale coating variations and improve process tolerance, enabling compatibility with cost-effective roll-to-roll (R2R) manufacturing[4–6]. Charge transport properties govern the performance of thick-film OSCs, as increased layer thickness amplifies the influence of three-dimensional nanomorphology on charge carrier dynamics. Thickening frequently induces morphological inhomogeneity, which perturbs the percolation pathways of bi-continuous networks and degrades charge carrier mobility[7–10]. Simultaneously, insufficient charge transport capacity exacerbates recombination losses in thick-film devices[11,12], imposing further efficiency limitations of OSCs. Therefore, understanding the structure-function relationship between molecular design and nanoscale phase separation, so as to improve the carrier transport property, is of vital significance for developing high-performance thick-film OSCs[3,13–15].

[1]School of Physics, State Key Laboratory of Crystal Materials, Shandong University, Jinan, Shandong, China. [2]School of Chemistry & Chemical Engineering, National Engineering Research Center for Colloidal Materials, Shandong University, Jinan, Shandong, China. [3]School of Chemistry and Environment, Heeger Beijing Research and Development Center, Beihang University, Beijing, China. [4]School of Chemistry and Chemical Engineering, Shandong Provincial Key Laboratory for Science of Material Creation and Energy Conversion, Science Center for Material Creation and Energy Conversion, Institute of Frontier Chemistry, Shandong University, Qingdao, Shandong, China. [5]Department of Chemistry, Hong Kong University of Science and Technology, Clear Water Bay, Kowloon, Hong Kong, China. ✉e-mail: ynsun@sdu.edu.cn; mjzhang@sdu.edu.cn; haoxt@sdu.edu.cn; hyin@sdu.edu.cn

In principle, high charge carrier mobility is a prerequisite for achieving high-performance thick-film OSCs[16]. Thickening of active layers inherently elongates charge transport pathways, thereby necessitating enhanced mobility to sustain sufficient diffusion lengths for effective carrier extraction at electrodes[17]. However, as shown in Supplementary Fig. 1, extracted charge carrier mobility values from prior studies reveal no statistically significant correlation between the performance of thick-film and thin-film OSCs. This observation fundamentally challenges the presumed theoretical feasibility of such thickness-mobility interdependence in the OSC field.

This phenomenon can be attributed to two major factors. First, the widespread methods for assessing mobility values predominantly depend on the zero-field mobility derived from the space-charge-limited current (SCLC) measurement[18,19]. However, the weak interactions among organic molecules, coupled with the blending of donor and acceptor materials within the active layer, inevitably introduce energetic disorder at charge carrier transport sites, resulting in a dependence on the global electric field[20]. Additionally, the work function difference between the anode and cathode creates a built-in electric field within the device, leading to deviations in carrier behavior from zero-field fitting values under operating conditions[21,22]. Even when field-dependent parameters, such as those governed by the Poole-Frenkel effect, are considered, a comprehensive understanding of charge carrier transport in OSCs remains elusive. This limitation arises from the distinctive blending characteristics of organic photovoltaic systems, which can induce multiple trap states and non-uniform electric field distributions[10,23]. Consequently, the effects of space-charge accumulation and carrier recombination dynamics introduce further uncertainties in the analysis of such field-dependent effects[24,25]. Therefore, developing robust analytical tools specifically tailored for thick-film OSCs is critical to resolving fundamental charge dynamics limitations and guiding practical manufacturing protocols.

In this work, we establish a critical-length-guided strategy for efficient screening of thick-film photovoltaic materials, enabling the fabrication of high-performance thick-film OSCs. First, we observe that the key factor determining the charge carrier transport property in thick-film systems is the critical length, which is a synergistic effect of zero-field mobility, hopping frequency, and the filed-dependent of mobility, rather than mobility alone. Furthermore, we establish a relationship between the critical length and the morphological property that enlarged acceptor domains in high critical length systems yield extended intermolecular hopping frequency, improved charge mobility, and reduced electric field dependence, collectively enabling optimal performance of thick-film OSCs. Through systematic screening of a series of photovoltaic materials, we identify that the bulk heterojunction (BHJ) system blended with the small molecular acceptor (SMA) BTP-eC9 exhibits a high critical length, which demonstrates a strong correlation with previously reported thick-film OSC performance in existing literature. Our investigations reveal that BTP-eC9 possesses a remarkable capability to significantly enhance thick-film device performance across diverse BHJ cases, establishing its critical role in advancing thick-film photovoltaic technologies, and a 19.0% efficiency is achieved in the thick-film D18:L8-BO:BTP-eC9 OSC. This work not only demonstrates the fabrication of high-performance thick-film OSCs, but also fundamentally advances material screening methodologies specifically tailored for thick-film-compatible organic semiconductors.

## Results

### Analysis of charge transport characteristics

To obtain the conductivity-frequency spectrum, we employed an electron-only device structure consisting of ITO/Al/active layer/PDINN/Ag, with the active layer thickness set at approximately 300 nm. The chemical structures of donor and acceptor materials used in this work are represented in Fig. 1a. As a commonly used donor in

OSCs, PM6 exhibits moderate energy levels and good compatibility with various acceptors, we employed PM6 as a uniform donor in our transport analysis experiments to systematically screen different acceptors. We also performed the charge carrier transport experiments using D18 as the donor material. As shown in Supplementary Figs. 2–3 and Supplementary Table 1, the critical length variation of D18-based active layers with the three acceptors exhibits a similar trend when compare to the PM6-based systems. Our analysis employed the alternating current (AC) conductivity framework, revealing two distinct conduction regimes[26,27]. In the low-frequency domain, conductivity remains constant and corresponds to the direct current (DC) conductivity. However, as the frequency of the applied alternating voltage increases progressively, the conductivity exhibits an exponential dependence on frequency[28–30], a characteristic behavior of the AC conductivity region. Papathanassiou et al.[31] explained the formation of AC conductivity in disordered solids by considering the distribution of charge carrier transport pathway lengths. Their study revealed that when the path length exceeds the critical length, charge carriers can move freely along the path. Conversely, if the charge carrier transport length falls below the critical length, carriers become localized at defect sites, thereby suppressing their transport dynamics. This confinement results in the observed exponential dependence of conductivity under high-frequency AC conditions. In the active layer of OSCs, the weak interactions among organic molecules cause a random distribution of conjugated lengths throughout the film, resulting in irregular distribution of transport sites. As a result, charge carrier transport pathways are not fixed but instead follow a distribution that aligns with the conditions of the model. The critical length represents the maximum distance a charge carrier can travel at an average velocity within a single hopping time[32]. As shown in Fig. 1b, a larger critical length signifies an increased capability for charge carriers to traverse longer distances within the film, which can enhance charge transport efficiency. By comparing the critical length, we screened acceptor materials suitable for thick-film OSCs. In active layers incorporating these acceptors, charge carriers can travel longer distances, improving charge transport efficiency. The critical length is determined using the following Eq. (1):

$$L_c = \frac{\mu_0}{2\beta^2 \omega_H} \tag{1}$$

where the $L_c$ is the critical length, $\mu_0$ is the zero-field mobility, $\beta$ is the filed-dependent of mobility and $\omega_H$ is the hopping frequency. The detailed derivation of Eq. (1) is provided in the ESI. Figure 1c depicts the relationship between critical length, mean velocity and hopping frequency. As shown in Fig. 2a, the hopping frequency of charge carriers can be extracted from the transition point between DC and AC conductivity using the Almond-West formula[33,34], as shown in Eq. (2):

$$\sigma(\omega) = \sigma_0 \left[ 1 + \left( \frac{\omega}{\omega_H} \right)^n \right] \tag{2}$$

$\sigma$ is the conductivity, $\sigma_0$ is the DC conductivity, $\omega$ is the frequency of the alternating electric field, and $n$ represents the frequency exponent in the high-frequency region of the conductivity.

Supplementary Table 2 presents the hopping frequency of carriers at different voltages across various systems. Those incorporating $PC_{71}BM$, L8-BO, and Y11 as acceptors exhibit the highest hopping frequencies of $1.09 \times 10^6$ rad/s, $6.41 \times 10^5$ rad/s, and $4.69 \times 10^5$ rad/s, respectively. In contrast, the ITIC-based film has the lowest hopping frequency of $2.56 \times 10^4$ rad/s among the evaluated systems. In the fabrication of thick-film OSCs, increasing the active layer thickness weakens the built-in electric field. Since charge carrier hopping in the active layer is field-dependent, a reduced field strength may prevent carriers from acquiring sufficient activation energy to reach the

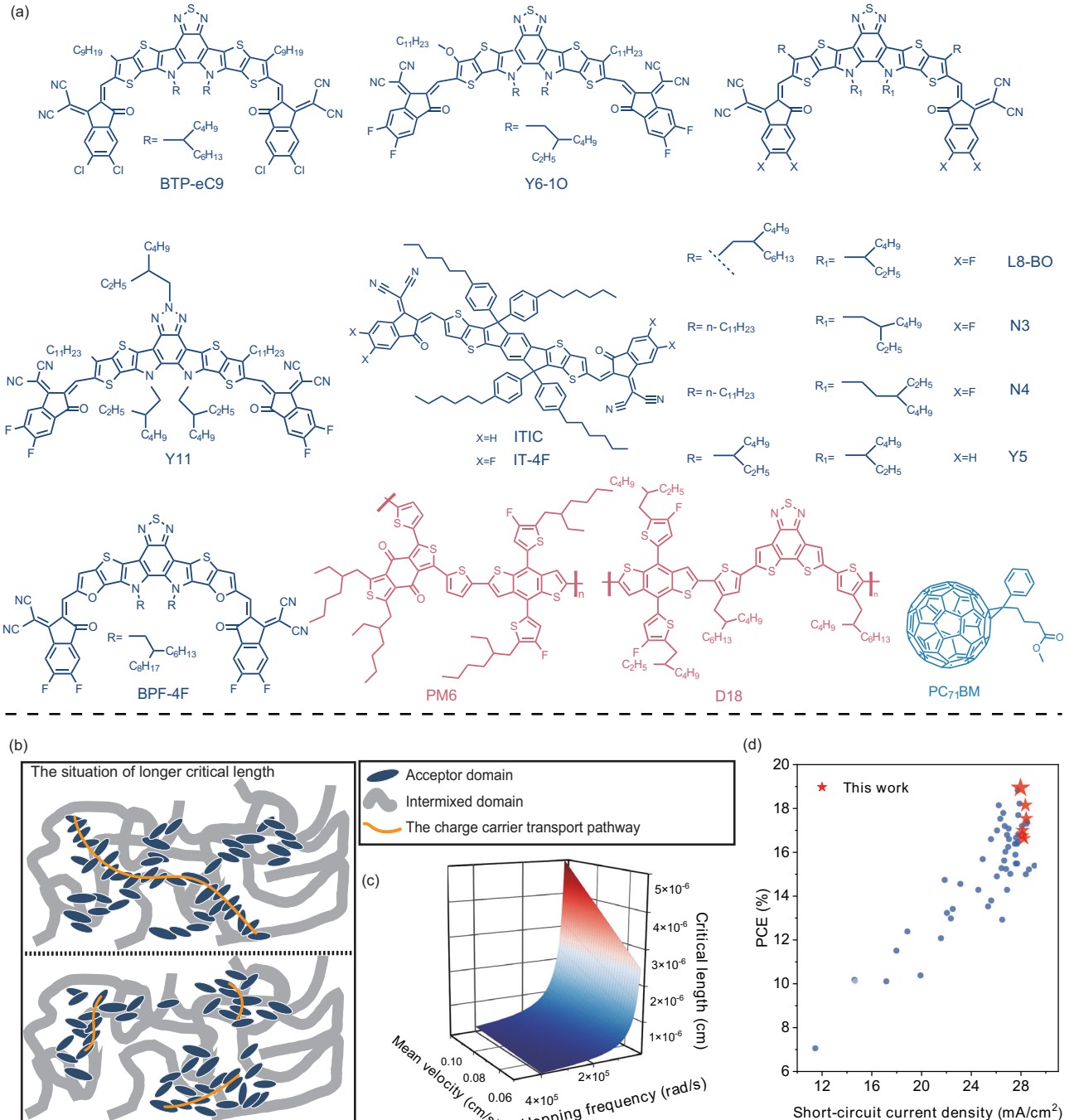

**Fig. 1 | Schematic diagram of the chemical structure of the material and AC conductivity model. a** Chemical structure of donors and acceptors. **b** A schematic representation of the model employed in the present study. **c** Schematic diagram of the relationship between critical length, mean velocity, and hopping frequency. **d** Plots of the PCE versus short-circuit current density for different systems compared to reported literature values.

transport energy level, thereby impeding the charge transport process and ultimately harming the device performance.

Based on this, we conducted measurements of the hopping frequency of charge carriers across various voltage levels to quantify hopping frequency variation in different systems. To precisely reflect the actual operating conditions of the device, we considered the built-in electric field of OSCs (~1 V) and its variation when the active layer thickness increases from 100 nm to 300 nm. Accordingly, we selected a voltage range of 1 V to 2 V to examine the relative change in hopping frequency Δ. The detailed description of Δ is summarized in the ESI, where Δ is a dimensionless parameter. Figure 2b shows the variation range of carrier hopping frequency in BTP-eC9 at 1 V and 2 V voltages.

The system composed of BTP-eC9 exhibits a Δ of 2.54. Figure 2c describes the voltage dependence of carrier hopping frequency across different systems. The red circles represent the absolute hopping frequencies of different systems, while the blue dots indicate the corresponding relative change, Δ. The gray connecting lines link the two parameters belonging to the same system. The AC conductivity spectra of the system used under different voltages and their variation with voltage are shown in Supplementary Figs. 4 and 5. The fullerene-based system using PC$_{71}$BM had a Δ value of 3.11. The highest Δ value of 16.51 is observed in the system composed with ITIC. A smaller voltage dependence of the hopping frequency indicates a reduced reliance of carrier hopping on external energy. As the active layer thickness

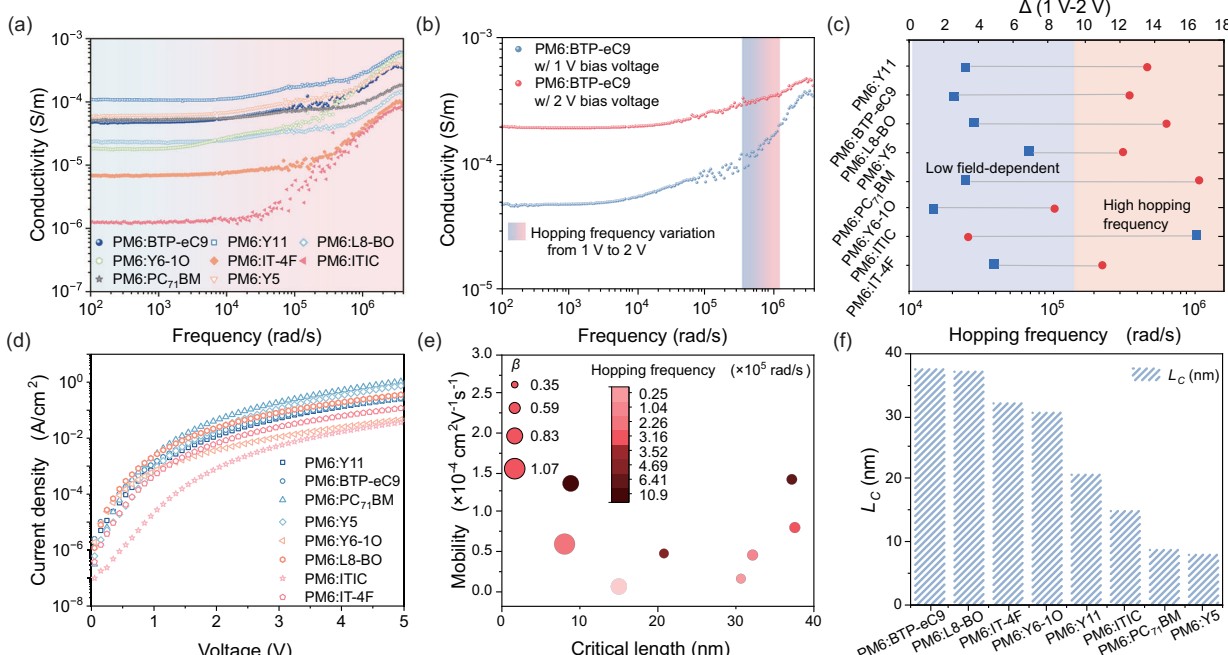

**Fig. 2 | Process diagram for calculating critical length. a** AC conductivity spectra of different systems. **b** AC conductivity of PM6:BTP-eC9 and the relative change of the hopping frequency on voltage within the range of 1 V to 2 V (Δ). **c** Hopping frequency of charge carriers in different systems and the relative change of the hopping frequency on voltage within the range of 1 V to 2 V. **d** The current density-voltage curve measured by SCLC. **e** The correlation between critical length and the zero-field mobility ($\mu_0$), filed-dependent of mobility ($\beta$), and hopping frequency ($\omega_H$). **f** The critical lengths of various systems. Source data are provided as a Source Data file.

increases, carrier transport becomes less sensitive to variations in electric field strength.

To further evaluate the potential for charge carrier transport over extended distances in active layers with different acceptor materials, we conducted a comparative analysis based on the concept of critical length as defined by the AC conductivity model. Determining the critical length requires assessing the average carrier velocity. We evaluated this using the space-charge-limited current (SCLC) method in an electron-only device[35]. Figure 2d presents the corresponding current density-voltage (J-V) curve. Due to the weak intermolecular interactions within the organic active layer and the resulting energetic disorder at transport sites, the charge carrier transport exhibit strong dependence on the electric field. Disregarding this dependence may lead to an inaccurate evaluation of the critical length. To account for this effect, we employed the Poole-Frenkel model[36] which allows the determination of both $\mu_0$ and $\beta$. Supplementary Table 3 summarizes the mobility values and their dependence on the electric field for different systems. Figure 2e presents a correlation analysis between critical length and zero-field mobility, along with $\beta$ and $\omega_H$. In the bubble plot, the color represents the hopping frequency, while the bubble size corresponds to the magnitude of $\beta$. The results reveal that a higher mobility does not necessarily correlate with a longer critical length. Instead, the charge carrier transport over extended distances within the active layer is not solely dictated by mobility but is instead influenced by three key factors: zero-field mobility, the field-dependent of mobility, and hopping frequency. The magnitude of a singular factor is insufficient to ascertain the critical length. Rather, an optimal combination of high mobility, low dependence on electric field strength, and elevated hopping frequency collectively contribute to an increased critical length. Consequently, the active layer demonstrates an enhanced capability for long-distance carrier transport.

The critical lengths of various systems, calculated based on the AC conductivity model, are presented in Fig. 2f. Among the evaluated systems, the BTP-eC9-based system exhibits the highest critical length.

The PM6:BTP-eC9 system has a mobility value of $8 \times 10^{-5}$ cm²V⁻¹s⁻¹ and a $\beta$ value of $5.5 \times 10^{-3}$ cm$^{1/2}$V$^{-1/2}$, resulting in a critical length value of 37.6 nm. Comparative analysis reveals distinct critical length characteristics among the studied systems. The PM6:L8-BO system shows critical lengths of 37.2 nm. The A-D-A type non-fullerene acceptor IT-4F exhibits a comparable value of 32.3 nm, securing the third position among the tested materials. In contrast, the fullerene acceptor PC$_{71}$BM exhibits a significantly smaller critical length of only 8.9 nm compared to these non-fullerene materials. A larger critical length indicates an enhanced ability of charge carriers to travel over extended distances within the active layer. The acceptor materials BTP-eC9, L8-BO, and IT-4F with larger critical lengths have the potential to manufacture more efficient thick-film OSCs. Based on this transport analysis, we implemented a ternary strategy by introducing BTP-eC9 (Y-series acceptor with maximum critical length) into the D18:L8-BO host system to achieve promising charge transport efficiency. This engineering approach yielded a critical length of 62.8 nm through optimized morphological control and energy level alignment, as systematically characterized in Supplementary Fig. 6 and Supplementary Table 3. Compared with binary D18:L8-BO and D18:BTP-eC9 systems, the ternary D18:L8-BO:BTP-eC9 system exhibits the largest critical length and achieves the highest PCE for 300 nm-thick OSCs.

The analysis above reveals that the Y-series acceptors BTP-eC9 and L8-BO exhibit the longest critical length. In practical applications, BTP-eC9 and L8-BO have been widely employed in the fabrication of thick-film OSCs, further demonstrating their effectiveness for enhancing charge transport and device performance. Zhang et al. optimized the crystal morphology of BTP-eC9 and achieved a power conversion efficiency (PCE) of 15.3% under treatment with halogen-free solvents[37], with an active layer thickness of 300 nm. Xue et al. attained a PCE of 15.6% in the PM6:BTP-eC9 system with an active layer thickness of 500 nm by extending the charge carrier transport distance[38]. Furthermore, Zhang et al. regulated the gradient distribution of donors and acceptors in the PM6:BTP-eC9 system, facilitating charge transfer

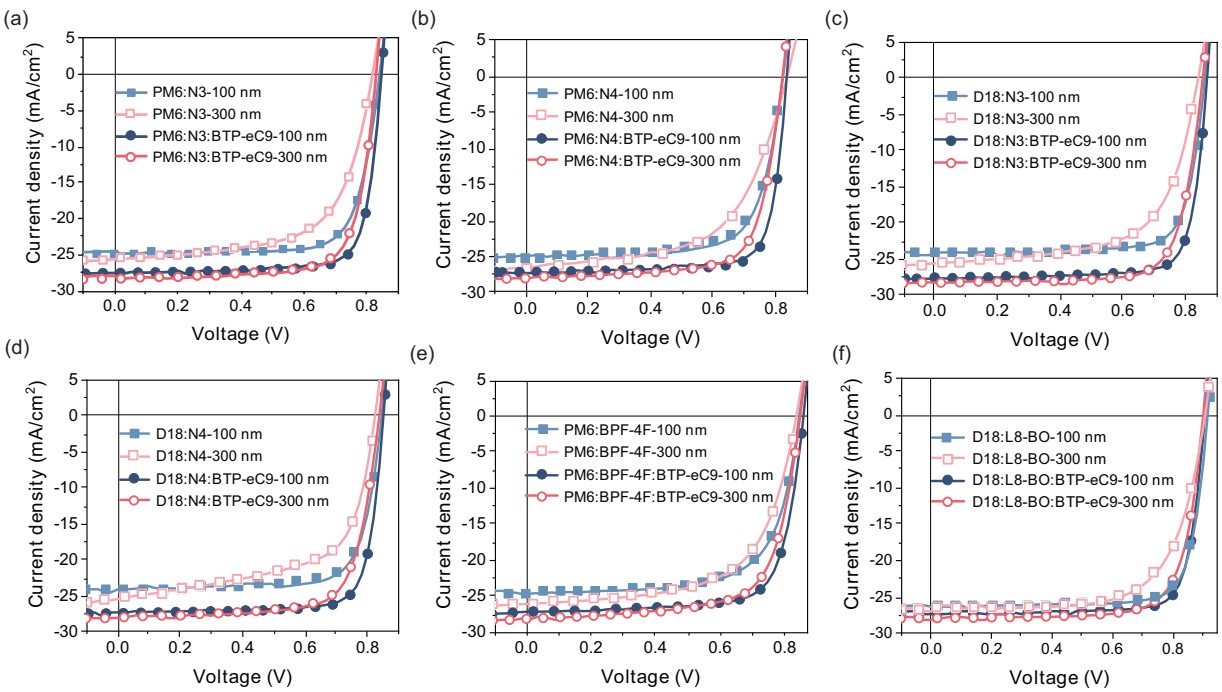

**Fig. 3 | Current density (*J*)-voltage (*V*) diagram. a** *J-V* diagram of PM6:N3 system. **b** *J-V* diagram of PM6:N4 system. **c** *J-V* diagram of D18:N3 system. **d** *J-V* diagram of D18:N4 system. **e** *J-V* diagram of PM6:BPF-4F system. **f** *J-V* diagram of D18:L8-BO system.

and achieving a remarkable PCE of 16.3% at a 300 nm active layer thickness[18]. Sun et al. used L8-BO as the acceptor to improve molecular stacking and extend the acceptor domain length in photovoltaic materials, achieving a high PCE of 17.2% in devices with an active layer with a thickness of 300 nm[39]. Zhang et al. implemented L8-BO as the acceptor and established a charge transfer channel through a buried layer structure, achieving a PCE of 16.0% at an active layer thickness of 500 nm[40]. The development of extended charge transfer pathways is crucial for thick-film OSCs, and the critical length provides a more precise assessment of carrier transport distances by incorporating both mobility and its electric field dependence. The magnitude of the critical length is intricately linked to the spatial extent of the acceptor domain within the active layer, offering a more comprehensive evaluation than mobility alone. In our experimental analysis, we conduct a comparative study of binary thick-film devices utilizing BTP-eC9 as the acceptor against other binary thick-film configurations. The findings indicate that binary devices employing BTP-eC9 with a larger critical length can achieve higher PCE values. Moreover, the widespread utilization of the BTP-eC9 and L8-BO, which exhibit greater critical lengths, in the fabrication of thick-film OSCs highlights the practical significance of utilizing critical length as a key criterion for selecting suitable acceptor materials in the optimization of thick-film OSCs.

### Device manufacturing

Based on the screening of the model, BTP-eC9 was introduced as the third component into different systems, and OSCs were fabricated with active layer thicknesses of 100 nm and 300 nm, respectively (Fig. 3). Table 1 provides a summary of the changes in device performance before and after the addition of BTP-eC9. The addition of BTP-eC9 resulted in an approximate 3.0% improvement in the power conversion efficiency (PCE) of the OSCs across all six systems, primarily due to enhancements in the short-circuit currents and fill factors (FFs). In OSCs featuring an active layer thickness of 300 nm, the incorporation of BTP-eC9 consistently enhances device efficiency compared to the control device without BTP-eC9, particularly in the D18:L8-BO system. After incorporating BTP-eC9 into the D18:L8-BO system, the

PCE of the solar cell with a 300 nm thick active layer reached 19.0%, marking one of the highest efficiencies reported for thick-film OSCs to date (Fig. 1d). The short-circuit current density reached 27.88 mA/cm². The EQE curve of the device with an active layer thickness of 100 nm is shown in Supplementary Fig. 7. After integrating BTP-eC9 into the D18:N3 system, the PCE of the solar cell with an active layer thickness of 300 nm achieved a remarkable PCE of 18.2%. The short-circuit current density value reached 28.27 mA/cm², while the value derived from the integration of the external quantum efficiency curve was 27.15 mA/cm² as shown in Supplementary Fig. 8. The large critical length of the active layer containing BTP-eC9 enhances the potential for long-range carrier transport and improves the ability of solar cells to convert light energy into electrical energy.

Femtosecond resolved transient absorption spectroscopy (TAS) was used to investigate the exciton dynamics and the hole-transfer dynamics in D18:L8-BO and D18:N3 systems after incorporating different content of BTP-eC9 with large critical lengths. We excited the acceptors in the blends using 800 nm excitation light to generate a photo-induced absorption band in the near-infrared wavelength region (NIR) and visible wavelength region. Figure 4a, b shows the TA spectra of D18:L8-BO and D18:L8-BO:8.3% BTP-eC9 at visible and NIR wavelengths, respectively. In the D18:L8-BO system, we found decay signals corresponding to localized excitons at about 860 nm (Fig. 4c). We employed bi-exponential decay fitting to obtain the average decay times for localized excitons. Figure 4d shows the variation of the average decay time of localized excitons with BTP-eC9 content, and the detailed average decay time values are shown in Supplementary Table 4. The average decay time of localized excitons in the D18:L8-BO system is 38.89 ps. Upon the addition of 8.3% BTP-eC9, this decay time decreases to 22.23 ps, indicating a significant reduction in localized exciton lifetime. As the BTP-eC9 content further increases, the average decay time of localized excitons initially increases, exceeding that of the system without BTP-eC9. However, when the BTP-eC9 content surpasses 83.3%, the average decay time decreases again. Only at a BTP-eC9 content of 8.3% does the average exciton decay time reach its minimum, indicating that optimal BTP-eC9 concentration can

**Table 1 | Summary of photovoltaic performance**

| | Thickness (nm) | $V_{OC}$ (V) | $J_{SC}$ (mA/cm²) | FF (%) | PCE (%) |
|---|---|---|---|---|---|
| D18:L8-BO | 100 | 0.92 | 26.33 | 78.4 | 18.9 (18.7 ± 0.2) |
| | 300 | 0.91 | 26.53 | 67.7 | 16.3 (16.0 ± 0.2) |
| D18:L8-BO:BTP-eC9 | 100 | 0.91 | 27.30 | 80.3 | 19.8 (19.6 ± 0.1) |
| | 300 | 0.90 | 27.88 | 75.4 | 19.0 (18.6 ± 0.2) |
| PM6:N3 | 100 | 0.84 | 24.83 | 75.2 | 15.7 (15.6 ± 0.3) |
| | 300 | 0.82 | 25.64 | 63.1 | 13.3 (13.1 ± 0.4) |
| PM6:N3:BTP-eC9 | 100 | 0.85 | 27.55 | 79.8 | 18.6 (18.5 ± 0.4) |
| | 300 | 0.83 | 28.34 | 74.8 | 17.6 (17.2 ± 0.3) |
| PM6:N4 | 100 | 0.82 | 25.12 | 69.7 | 14.4 (14.1 ± 0.4) |
| | 300 | 0.80 | 26.67 | 56.6 | 12.1 (12.0 ± 0.3) |
| PM6:N4:BTP-eC9 | 100 | 0.84 | 27.14 | 79.1 | 17.9 (17.7 ± 0.4) |
| | 300 | 0.82 | 28.02 | 72.9 | 16.8 (16.5 ± 0.2) |
| PM6:BPF-4F | 100 | 0.85 | 24.52 | 68.1 | 14.1 (13.9 ± 0.3) |
| | 300 | 0.82 | 25.61 | 57.3 | 12.1 (11.7 ± 0.4) |
| PM6:BPF-4F:BTP-eC9 | 100 | 0.86 | 27.19 | 76.2 | 17.8 (17.5 ± 0.4) |
| | 300 | 0.85 | 28.16 | 69.6 | 16.6 (16.4 ± 0.4) |
| D18:N4 | 100 | 0.84 | 24.09 | 73.9 | 15.0 (14.8 ± 0.5) |
| | 300 | 0.82 | 25.43 | 61.2 | 12.8 (12.5 ± 0.4) |
| D18:N4:BTP-eC9 | 100 | 0.85 | 27.41 | 79.5 | 18.5 (18.2 ± 0.5) |
| | 300 | 0.84 | 28.05 | 72.4 | 17.0 (16.8 ± 0.2) |
| D18:N3 | 100 | 0.86 | 24.17 | 76.9 | 16.0 (15.9 ± 0.3) |
| | 300 | 0.84 | 25.62 | 64.2 | 13.8 (13.5 ± 0.3) |
| D18:N3:BTP-eC9 | 100 | 0.87 | 27.61 | 80.9 | 19.4 (19.1 ± 0.3) |
| | 300 | 0.86 | 28.27 | 75.1 | 18.2 (17.9 ± 0.2) |

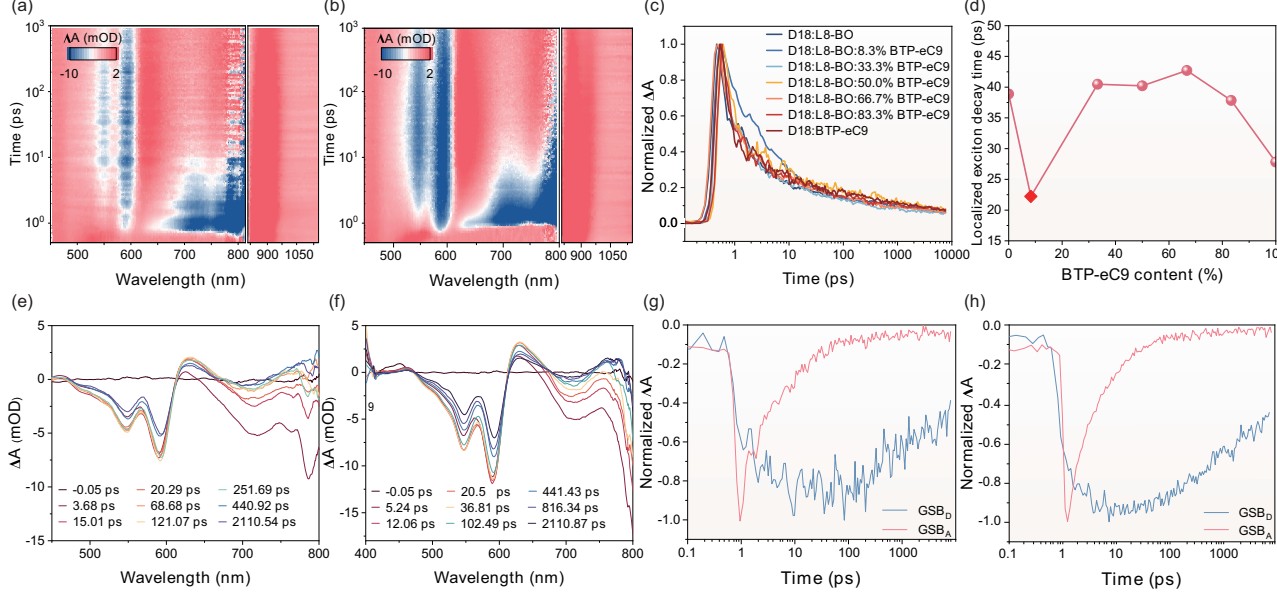

**Fig. 4 | Femtosecond resolved transient absorption spectroscopy (TAS) results. a** 2D TA data of D18:L8-BO. **b** 2D TA data of D18:L8-BO:8.3% BTP-eC9. **c** Localized exciton decay signals extracted from 860 nm. **d** Variation of average decay time of localized exciton with BTP-eC9 content. **e** TA spectra at different delay times of D18:L8-BO. **f** TA spectra at different delay times of D18:L8-BO:8.3% BTP-eC9. **g** Kinetic traces at the selected wavelength of D18:L8-BO (The ground-state bleaching (GSB) signal of the donor is located at 590 nm). **h** Kinetic traces at the selected wavelength of D18:L8-BO:8.3% BTP-eC9.

accelerate the decay of localized excitons. For the D18:N3 system, localized and delocalized excitation signals appeared at about 912 nm and 1432 nm, respectively (Supplementary Figs. 9–14, Supplementary Table 5 and Supplementary Table 6). The curves of average decay time of localized excitation signals and delocalized excitation signals as a function of BTP-eC9 content were obtained by three exponential decay fitting, as shown in Supplementary Fig. 9g, Supplementary Fig. 9h, Supplementary Tables 5 and 6. At a BTP-eC9 content of 30%, the decay of localized excitons was observed to occur with the shortest decay time. This accelerated decay facilitates exciton dissociation, thereby enhancing charge generation[41,42]. In the visible wavelength range, we further investigated the effect of 8.3% BTP-eC9 content on

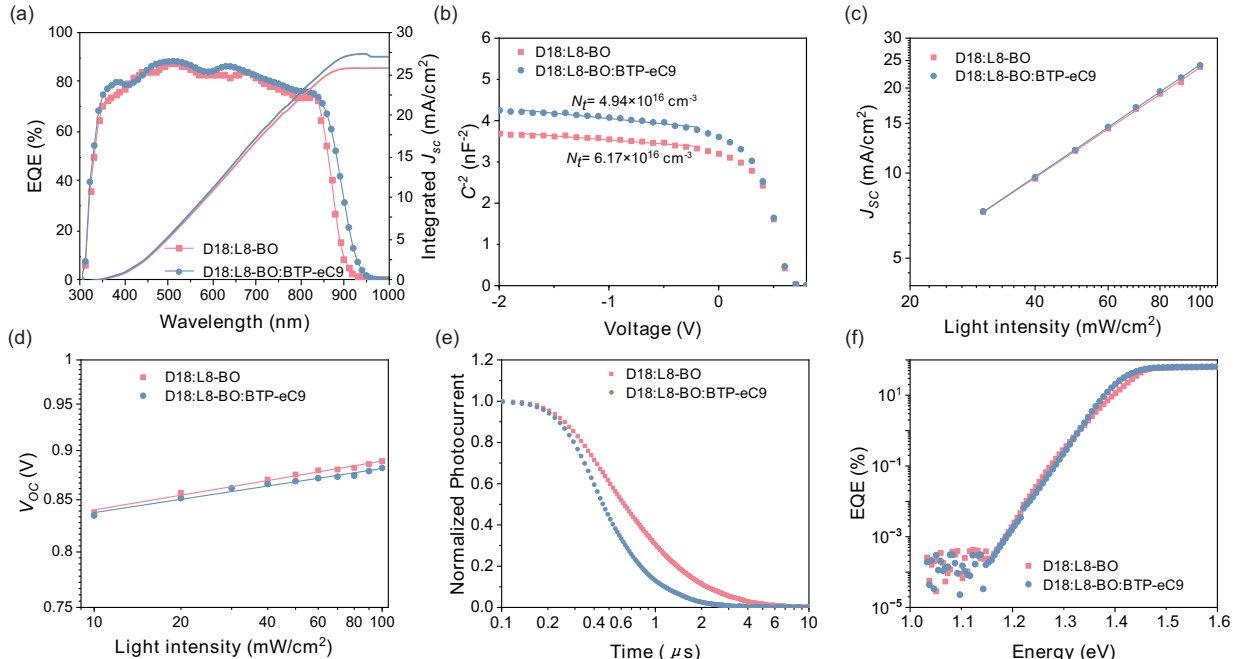

**Fig. 5 | Charge extraction, recombination, and defect state analysis of D18:L8-BO system with 300 nm-thick active layer. a** EQE spectra of devices based on D18:L8-BO system. **b** Extracted $C^{-2}$ versus $V$ plots at 10 kHz, $N_t$ is the defect state density. **c** The dependence of short-circuit current density on light intensity. **d** The dependence of open-circuit voltage on light intensity. **e** Normalized TPC data for the devices based on D18:L8-BO system. **f** FTPS data for the devices based on D18:L8-BO system.

hole transfer. As shown in Fig. 4e, f and Fig. 4f, the ground state bleaching (GSB) signal of the donor was observed around 590 nm. By bi-exponential fitting (Fig. 4g, h, and Supplementary Table 7), the hole transfer rate increased from 0.80 ps$^{-1}$ to 1.27 ps$^{-1}$. This indicates that the addition of 8.3% BTP-eC9 accelerates hole transfer, reduces the probability of electron-hole recombination, and contributes to the enhancement of the photogenerated current.

Figure 5a shows the external quantum efficiency (EQE) curve of a device based on the D18:L8-BO:BTP-eC9 system, with the integrated $J_{SC}$ value calculated as 27.34 mA/cm². The presence of defect states and carrier recombination are critical factors that degrade the performance of thick-film devices. To investigate variations in defect state density ($N_t$), we performed capacitance ($C$) versus voltage ($V$) measurements[43] of D18:N3 and D18:L8-BO systems under dark conditions. The defect state density was quantified by analyzing the slope of the $1/C^2$-$V$ plot. Figure 5b shows the results of defect state analysis of D18:L8-BO:BTP-eC9 systems. The findings reveal that incorporating BTP-eC9 into the active layer reduces the defect state density from $6.17 \times 10^{16}$ cm$^{-3}$ to $4.94 \times 10^{16}$ cm$^{-3}$, indicating an improvement in film quality and reduced trap-assisted recombination. As shown in Supplementary Fig. 8 and Supplementary Table 8, for D18:N3 system, the defect density of states decreased from $1.82 \times 10^{17}$ cm$^{-3}$ to $1.56 \times 10^{17}$ cm$^{-3}$ due to the addition of BTP-eC9. The incorporation of BTP-eC9 reduces the capture of charge carriers by defects during transport, resulting in an increased collection of charge carriers by the electrode, thereby improving the device performance. A reduction in defect density means fewer trap states, which prolongs carrier lifetime and reduces the recombination probability, allowing carriers to travel longer distances before being trapped[44]. In the subsequent study of device recombination and carrier lifetimes, the incorporation of BTP-eC9 was found to suppress recombination and extend carrier lifetimes, which is consistent with the observed reduction in defect states.

Subsequently, we investigated the charge carrier recombination within the device by analyzing the relationship between $J_{SC}$ and light intensity ($P_{light}$), characterized by $J_{SC} \propto P_{light}^{\alpha}$, where $\alpha$ varies between

0 and 1[45]. Fig. 5c illustrates the dependence of short-circuit current density on light intensity of D18:L8-BO system. In OSCs with an active layer thickness of 300 nm, the incorporation of BTP-eC9 led to an increase in the $\alpha$ from 0.97 to 0.99. This increase suggests that the addition of BTP-eC9 suppresses bimolecular recombination within the device, thereby contributing to an improvement in both the FF and short-circuit current density. For the D18:N3 system, the value of $\alpha$ is consistently 0.99 (Supplementary Fig. 15 and Supplementary Table 9), indicating that the bimolecular recombination is not dominant in such cases. The dependence of open-circuit voltage on light intensity follows the relationship $V_{OC} \propto \frac{nkT}{q} \log P_{light}$ [46,47], where $V_{OC}$ is the open-circuit voltage, $k$ is the Boltzmann constant, $T$ is the temperature and $q$ is the elementary charge. By doping BTP-eC9 into the 300-nm D18:L8-BO film, $n$ values were reduced from 1.89 to 1.70, indicating that BTP-eC9 doping effectively suppressed the trap-assisted recombination within the device (Fig. 5d). For D18:N3 system (Supplementary Fig. 8 and Supplementary Table 10), by doping BTP-eC9 into the 300-nm active layer, $n$ values were reduced to 1.87 and 1.64, indicating that the addition of BTP-eC9 reduces trap-assisted recombination within the device. This suppression is likely associated with the reduction of defect states, which in turn minimizes the charge carrier trapping properties.

Figure 5e illustrates the variation in transient photocurrent over time of D18:L8-BO system. A fast carrier extraction time is essential for charge recombination probability, while a prolonged carrier lifetime is crucial for enabling long-range carrier transport in thick-film devices. Transient photocurrent (TPC) measurements revealed that the extraction time of charge carriers decreased from 0.93 $\mu$s to 0.43 $\mu$s after doping with BTP-eC9 for devices with a 300 nm active layer. Specifically, for the D18:N3 system (Supplementary Fig. 8e and Supplementary Table 11), the extraction time of charge carriers decreased from 0.28 $\mu$s to 0.24 $\mu$s after doping with BTP-eC9 for devices with a 300 nm active layer. This reduction in extraction time mitigates the probability of charge carriers being captured and released by trap states, enabling more efficient charge transport to the corresponding

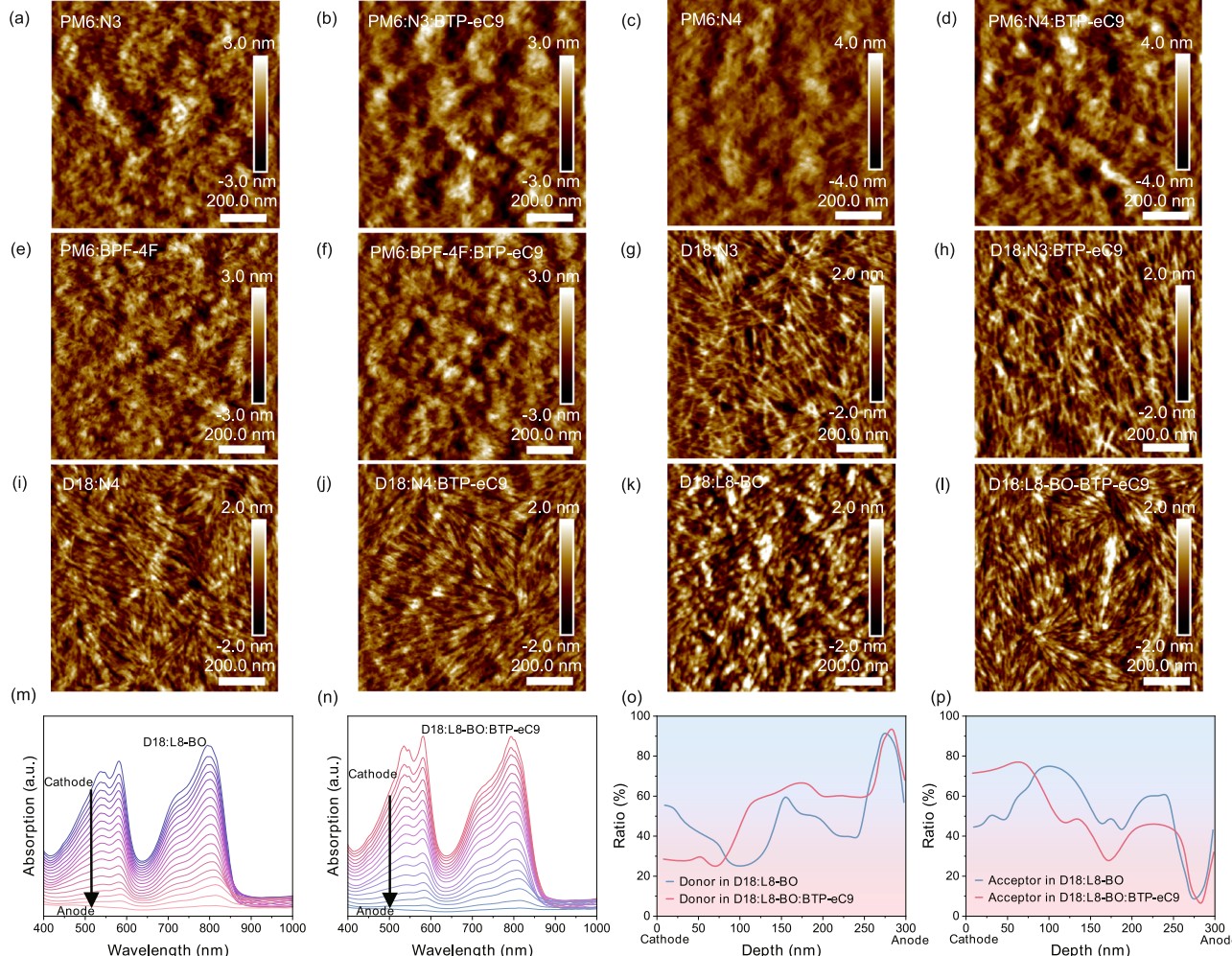

**Fig. 6 | Aggregation state and vertical phase analysis. a** AFM image of PM6:N3. **b** AFM image of PM6:N3:BTP-eC9. **c** AFM image of PM6:N4. **d** AFM image of PM6:N4:BTP-eC9. **e** AFM image of PM6:BPF-4F. **f** AFM image of PM6:BPF-4F:BTP-eC9. **g** AFM image of D18:N3. **h** AFM image of D18:N3:BTP-eC9. **i** AFM image of D18:N4. **j** AFM image of D18:N4:BTP-eC9. **k** AFM image of D18:L8-BO. **l** AFM image of D18:L8-BO:BTP-eC9. **m** D18:L8-BO film-depth-dependent absorption spectrum. **n** D18:L8-BO film-depth-dependent absorption spectrum. **o** Distribution of donors in the active layer in the vertical direction. **p** Distribution of acceptors in the active layer in the vertical direction.

electrode. Concurrently, we performed transient photovoltaic (TPV) testing on the device to investigate variations in carrier lifetime. The test results in Supplementary Fig. 16, Supplementary Fig. 8 and Supplementary Table 12 indicate that the incorporation of BTP-eC9 extends the carrier lifetime, thereby demonstrating that the suppression of trap-assisted recombination enhances the carrier lifetime. This enhancement increases the probability of carriers being collected by the electrode, thereby facilitating current generation.

The decrease in the density of states within the bandgap is beneficial for reducing scattering and trapping during charge transport, thereby improving the efficiency of charge transport[48]. The Urbach energy, measured by Fourier transform photocurrent spectroscopy (FTPS), was used to represent the size of the density of states within the bandgap. For the D18:L8-BO system, when the thickness of active layer was 300 nm, the highest Urbach energy of the device was 23.0 meV. After adding BTP-eC9, the Urbach energy of the device with the 300-nm active layer decreased to 20.8 meV (Fig. 5f). While for the D18:N3 system, the Urbach energy in 300-nm devices decreased from 26.5 meV to 24.8 meV after adding BTP-eC9, respectively (Supplementary Fig. 8 and Supplementary Table 13). The decrease in Urbach energy after adding BTP-eC9 indicates a reduction in the density of defect states within the bandgap, leading to a narrower tail in the density of states and a flatter transport energy landscape. The

reduction in defect state density within the bandgap helps to reduce the capture of charge carriers during the transport process. This is particularly important for thick-film OSCs, as they require longer charge carrier transport pathways to sustain high FF values.

## Morphological analysis

To investigate the factors responsible for the high hopping frequency and critical length of the active layer formed by BTP-eC9, we incorporated BTP-eC9 into various BHJ systems, including PM6:N3, PM6:N4, PM6:BPF-4F, D18:N3, D18:N4 and D18:L8-BO films. These modified systems were subsequently analyzed using atomic force microscopy (AFM) and grazing incidence small-angle X-ray scattering (GISAXS) to evaluate morphological and structural changes. Figure 6a–l display the AFM images of 300-nm active layers. The findings indicate that the incorporation of BTP-eC9 into binary systems promotes the formation of continuous and elongated aggregation patterns within both the PM6 and D18 based systems. Compared to scenarios where donor and acceptor materials are isolated and randomly dispersed, the presence of a continuous and extensive aggregation pattern facilitates long-range charge carrier transport. This morphological feature enables carriers to traverse the active layer more efficiently, reach the corresponding electrode and be collected[49]. Similarly, Supplementary Fig. 17 presents the AFM images of the active layer with a thickness of

100 nm where a comparable phenomenon can be observed. The incorporation of BTP-eC9 induced an elongated aggregation morphology, even in thinner active layers. The elongated aggregation morphology of the BTP-eC9 system contributes to an increased critical length, thereby validating the appropriateness of employing the critical length model for the selection of acceptor materials suitable for thick-film OSCs. To examine the impact of BTP-eC9 on the vertical distribution of donor and acceptor materials, we conducted film-depth-dependent light absorption spectroscopy (FLAS) measurements for the D18:L8-BO and D18:N3 systems. The results of D18:N3 system are shown in Supplementary Figs. 18–20. Figure 6m, n shows the FLAS spectra of D18:L8-BO and D18:L8-BO:8.3% BTP-eC9, respectively. We extracted the distribution of donors and acceptors in the active layer. The FLAS spectra in D18: L8-BO with different proportions of BTP-eC9 are shown in Supplementary Figs. 21 and 22, and the corresponding distribution of donors and acceptors are shown in Supplementary Figs. 23 and 24. The results revealed that the incorporation of BTP-eC9 led to an increased donor concentration near the anode and a higher acceptor concentration near the cathode (Fig. 6o, p). This optimized vertical phase distribution can effectively reduce the charge carrier recombination near the electrodes and facilitate the charge transport process.

We further employed GISAXS analysis to examine variations in the average size of the acceptor domain before and after introducing BTP-eC9. Figure 7a–l shows the GISAXS images of different systems. The GISAXS images with an active layer thickness of 100 nm are shown in Supplementary Fig. 25. By extracting one-dimensional line-cut profiles from the corresponding two-dimensional GISAXS images, we applied fractal fitting to determine the acceptor domain size in the high $q$-value region[50]. The fitting process is shown in Supplementary Figs. 26–29. The incorporation of BTP-eC9 results in an increase in the average size of the acceptor domain within the active layer. Figure 7m–n depict the changes in domain size after the addition of BTP-eC9, showing that in the PM6:BPF-4F system with a 100 nm-thick active layer, the acceptor domain size increased from 35 nm to 48 nm after the addition of BTP-eC9. Similarly, in the D18:N3 and D18:N4 systems, the acceptor domain sizes grew from 24 nm and 20 nm to 27 nm and 29 nm, respectively. Notably, for the D18:N3 and D18:L8-BO systems with a 300 nm-thick active layer, the acceptor domain sizes increased from 22 nm and 16 nm to 27 nm and 20 nm, respectively. Detailed data on the average acceptor domain size and intermixing domain size can be found in Supplementary Tables 14–17. The increase of the acceptor domain size promotes more efficient transport of charge carriers, thereby extending the average distance over which charge carriers move within the acceptor domain. Morphological evolution analysis reveals that acceptor domain enlargement enhances charge transport dynamics through two synergistic mechanisms: (1) establishing extended percolation pathways that increase the mean free path of charge carriers within acceptor phases, and (2) reducing trap-assisted recombination probability. This improvement creates favorable conditions for achieving high-performance thick-film OSCs, especially the FF values.

## Discussion

This work proposes a framework for acceptor screening in thick-film OSCs through the systematic integration of charge carrier hopping dynamics and field-dependent transport behavior. The model establishes the critical length as a significant descriptor for quantifying long-range charge carrier transport capabilities under operational conditions. Rational selection criteria based on critical length parameters, when applied to diverse binary systems, demonstrate substantial enhancements in both thin-film and thick-film device performance. Optimized architectures incorporating this methodology achieve desirable device performance, with the D18:L8-BO:BTP-eC9 system exhibiting a champion PCE of 19.0% in the 300-nm thick-film OSC. This methodology systematically integrates the charge carrier transport

models, spectroscopic models of charge carrier dynamics, and device performance, thereby positioning critical length analysis as a cornerstone for next-generation OSC design. The derived structure-property-performance correlations advance fundamental understanding while providing actionable guidelines for industrial translation of high-throughput OSC manufacturing.

## Methods

### Materials

PM6, D18, Y11, BTP-eC9, L8-BO, Y5, Y6-1O, ITIC, IT-4F, N3, N4 were purchased from Solarmer Materials Inc and eFlexPV Limited. PEDOT:PSS (CLEVIOSTM PVP AI 4083, Heraeus, Germany) was purchased from Xi'an Polymer Light Technology Corp. The indium tin oxide (ITO) substrates were purchased from Advanced Election Technology CO,.Ltd. All materials were used as received without further purification.

### Electron-only devices fabrication and characterization

The electron-only devices were fabricated with a conventional structure of ITO/Al (50 nm)/BHJ (300 nm)/PDINN/Ag. The patterned indium tin oxide (ITO)-coated substrates were placed in an ultrasonic bath and cleaned sequentially with detergent, deionized water, acetone, anhydrous ethanol, and isopropanol for at least 20 min each time. Then, the patterned indium tin oxide (ITO)-coated substrate was irradiated with UV-ozone for 15 min to enhance its wettability with the solution. Subsequently, a layer of 50 nm Al was deposited on the ITO substrates under vacuum to block the transport of hole carriers. Subsequently, the substrate with the deposited Al barrier layer was transferred to a glove box filled with nitrogen for spin coating of the active layer. The active layer solution is obtained by mixing donors and acceptors in different ratios, then dissolving them in chloroform solvent and heating and stirring for 2 h. Half an hour before spin coating the active layer, an additive such as 1,8-Diiodeoctane (DIO) was added to the stirred solution. When spin coating the active layer, the speed of the spin coating machine is 3000 rpm/s and the acceleration is 2000 rpm/s$^2$. The substrate coated with the active layer was annealed on a hot stage at 100 °C for 10 min. After cooling, the electron transport layer PDINN was spin coated. Finally, a layer of 100 nm thick Ag was deposited as the cathode on the topmost layer of the substrate after spin coating with PDINN in a vacuum state. The active device area was 0.024 cm$^2$. The current density-voltage ($J$-$V$) characteristic curves of all devices were recorded in a low vacuum environment by employing a computer-controlled Keithley 2612B. The measurement of AC conductivity is carried out in HIOKI 9259-10 under low vacuum conditions.

### Fabrication of organic solar cells

The structure of the organic solar cell is ITO/PEDOT:PSS/active layer/PDINN/Ag. The patterned indium tin oxide (ITO)-coated substrates were placed in an ultrasonic bath and cleaned sequentially with detergent, deionized water, acetone, anhydrous ethanol, and isopropanol for at least 20 min each time. The cleaned ITO substrate was exposed to UV-ozone for 15 min, followed by spin-coating of the hole transport layer, PEDOT:PSS. Thermal annealing was performed at 150 °C for 15 min, they were then transferred to a nitrogen-filled glovebox. The donor and acceptor materials in all systems were dissolved in chloroform (CF) and heated with stirring for two hours. For the PM6-based system, 0.5% 1,8-Diiodeoctane (DIO) was added as an additive 30 min prior to spin-coating the active layer. After spin-coating, the PM6-based system underwent thermal annealing at 100 °C for 10 min, while the D18-based system was subjected to chloroform vapor treatment for 1 min or use additive 1,3,5-trichlorobenzene. After cooling, the electron transport layer PDINN was spin coated. Finally, a layer of 100 nm-thick Ag was deposited as the cathode on the topmost layer of the substrate after spin coating with PDINN in a vacuum state. In the D18:L8-BO:BTP-eC9 system, the total donor-to-acceptor ratio is 1:1.2,

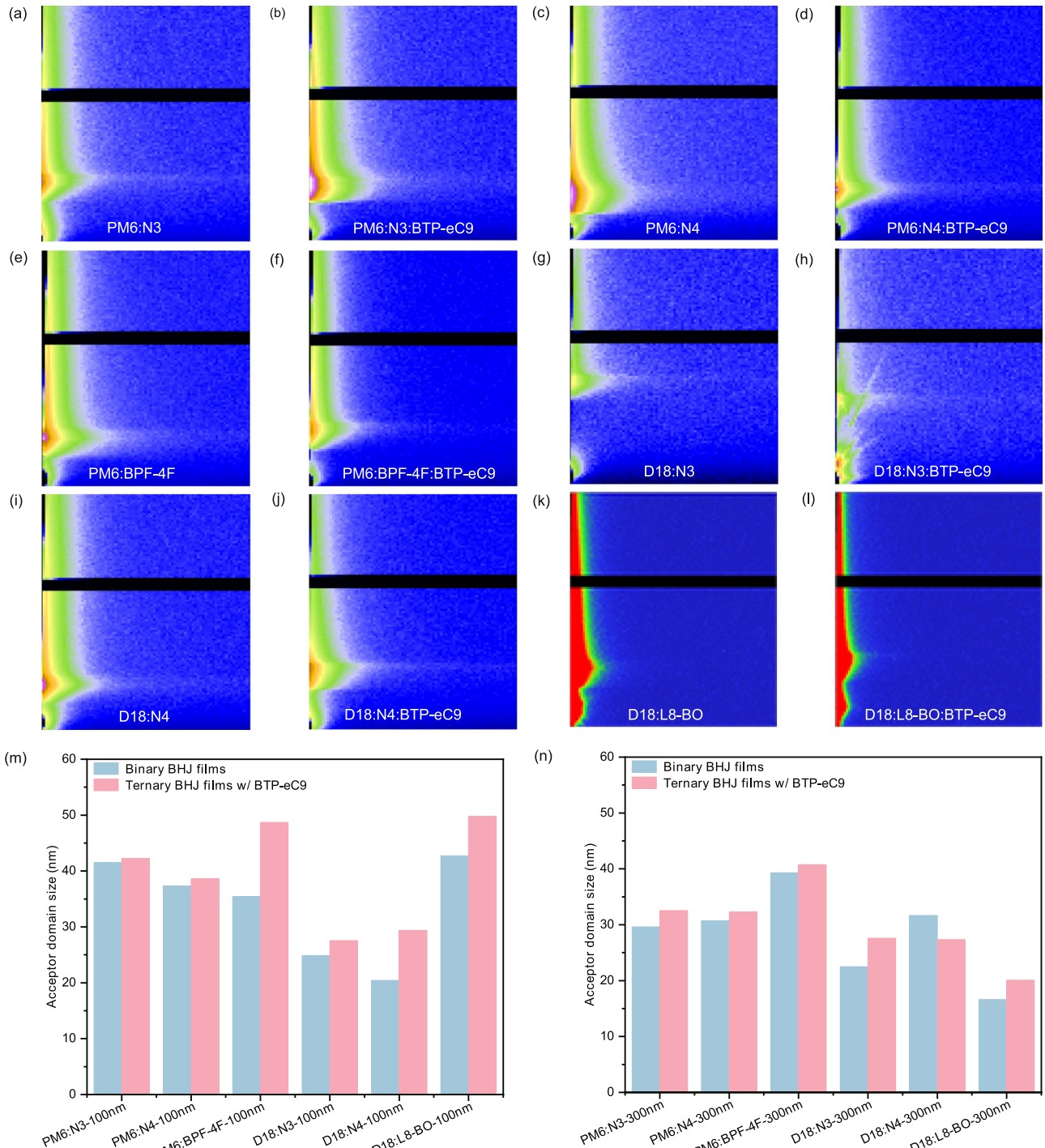

**Fig. 7 | Acceptor domain size analysis. a** GISAXS image of PM6:N3. **b** GISAXS image of PM6:N3:BTP-eC9. **c** GISAXS image of PM6:N4. **d** GISAXS image of PM6:N4:BTP-eC9. **e** GISAXS image of PM6:BPF-4F. **f** GISAXS image of PM6:BPF-4F:BTP-eC9. **g** GISAXS image of D18:N3. **h** GISAXS image of D18:N3:BTP-eC9. **i** GISAXS image of D18:N4. **j** GISAXS image of D18:N4:BTP-eC9. **k** GISAXS image of D18:L8-BO. **l** GISAXS image of D18:L8-BO: BTP-eC9. **m** Changes in the acceptor domain size after adding BTP-eC9 to active layer with a thickness of 100 nm. **n** Changes in the acceptor domain size after adding BTP-eC9 to active layer with a thickness of 300 nm.

with the acceptor components L8-BO and BTP-eC9 mixed at a ratio of 1.1:1. In the D18:N3:BTP-eC9 system, the overall donor-to-acceptor ratio is also 1:1.6, where the acceptor blend ratio of N3 to BTP-eC9 is 1.12:0.48.

## Atomic force microscope (AFM) images

AFM measurements were performed on a Dimension Icon AFM (Bruker) in a tapping mode under ambient conditions.

## GISAXS measurements

GISAXS measurements were performed at the Shanghai Synchrotron Radiation Facility BL16B1 beamline under ambient conditions and a Xeuss 3.0 SAXS/WAXS laboratory beamline at Vacuum Interconnected Nanotech Workstation (Nano-X) in China with K$\alpha$ X-ray of Cu source (operated at 50 kV, 0.6 mA, 1.5419 Å). X-rays have a wavelength of 1.23984 Å and sample detector distance was calibrated by a silver behenate (AgBH). The angles of grazing incidence for GISAXS

measurements are 0.15°. Samples were prepared on silicon/poly(3,4-ethylenedioxythiophene) polystyrene sulfonate substrates using identical blend solutions and methods as those used in photovoltaic device fabrication.

### Film-depth-dependent light absorption spectroscopy (FLAS)

The depth-dependent optical absorption spectra were obtained using a film-depth-dependent optical absorption spectrometer (PU100, Puguangweishi Co. Ltd). In-situ soft plasma etching under low pressure (less than 20 Pa) was used to extract the depth-resolved absorption spectra of the organic active layer. The FLAS results were fitted using the Beer-Lambert law, and the exciton generation profile was subsequently fitted using an improved optical matrix-transfer method.

### Transient absorption (TA) spectroscopy

Femtosecond TA spectroscopy measurements were conducted using an Ultrafast Helios pump-probe system in combination with a regenerative amplified laser setup. A Ti:sapphire amplifier (Astrella, Coherent) generated an 800 nm pulse with a 100 fs pulse width, 7 mJ/pulse energy, and a repetition rate of 1 kHz. This pulse was split into two parts: one was directed into an optical parametric amplifier (TOPAS, Coherent) to produce 800 nm pump pulses, while the other was focused onto sapphire and YAG plates to generate a broad probe light spectrum ranging from 400 to 1200 nm. The time delay between the pump and probe pulses was adjusted using a motorized optical delay line with a maximum delay of 8 ns. The samples were fabricated on quartz substrates and encapsulated in a nitrogen-filled glovebox using epoxy resin to reduce air-induced degradation.

### Reporting summary

Further information on research design is available in the Nature Portfolio Reporting Summary linked to this article.

## Data availability

The data that support the findings of this study are available within the article and its Supplementary Information/Source Data file. Source data are provided with this paper.

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

## Acknowledgements

This work was supported by the National Natural Science Foundation of China (grant nos. 12204272 (H.Yin.)), Major Program of Natural Science Foundation of Shandong Province (grant nos. ZR2019ZD43 (H.Yin.)). M. Zhang and X. Guo acknowledge financial support from Shandong Provincial Natural Science Foundation (ZR2022JQ09 (M. Zhang)), Taishan Scholar Program at Shandong Province (tsqn202306061 (M. Zhang)). The authors thank the beamline BL16B1 of Shanghai Synchrotron Radiation Facility for supporting the GISAXS measurements and beam time and technical support provided by in-house X-ray scattering beamline of National Engineering Research Center for Colloidal Materials, Shandong University.

## Author contributions

H. Yin conceived the ideas and visualization. Y.M. conducted research on carrier transport characteristics, device measurements, data collection and organization, wrote the original draft and visualization. B.C., Yanna Sun, fabricated devices and EQE measurements. J.S., D.J., B.S., R.G. M.Z. contributed to the data analysis about the charge carrier transport properties. D.J. and J.S. performed the GISAXS measurements. H.M. performed the AFM measurements. J.Q. and P.L. performed the TA measurements. J.Z., L.W., X.D., X.G., K.G., H. Yan, M.Z., F.C., Yanming Sun, and X.H. contributed to the research preparation and data analysis. All authors contributed to discussions, manuscript writing, and revisions.

## Competing interests

The authors declare no competing interests.
