## [Transparent Peer Review file · Nature Communications]

Critical Length Screening Enables 19% Efficiency in Thick-Film Organic Solar Cells

Corresponding Author: Professor Hang Yin

Version 0:

Reviewer comments:

Reviewer #1

(Remarks to the Author)

This manuscript investigates the critical length in organic photovoltaic (OPV) devices, analyzing its dependence on three key parameters: mobility, field-dependent mobility, and hopping frequency. The experimental results reveal that higher mobility does not always correlate with longer critical length. Instead, the critical length is determined by the interplay of these three factors. Among the studied systems, PM6 paired with BTP-eC9 or L8-BO exhibits a longer critical length compared to other acceptors (ITIC, Y5, and PCBM). Notably, incorporating BTP-eC9 into the D18:L8-BO system further extended the critical length, enabling a power conversion efficiency (PCE) of 19% in a 300 nm-thick active layer device. The study underscores the significance of critical length in OPVs, particularly for thick-film devices relevant to commercialization. However, the manuscript's current presentation lacks clarity in some areas, and certain analyses are insufficient to fully support the conclusions. Due to these issues, a comprehensive assessment is challenging at this stage. I recommend revising the manuscript to improve data presentation and analytical rigor before further evaluation. A second review may be necessary once additional details are provided.

1. In Figure 1c, which material system does the data represent? Is this trend consistent across other systems studied?
2. The manuscript mentions using SCLC method to determine average carrier velocity. However, equation (1) and Poole-Frenkel model do not explicitly involve carrier velocity. Using zero field mobility and electric field intensity, one can determine the critical length without knowing the average carrier velocity. So, I don't get the point why average carrier velocity was brought in this analysis. Furthermore, according to Ref 37 and my knowledge, SCLC method can provide the mobility only and it is unclear how the carrier velocity was determined. Perhaps, authors used the relationship between carrier drift velocity and mobility ($v = \mu E$), but it should be clearly stated. Additionally, since velocity is field-dependent, it may vary with film thickness—was this accounted for?
3. Perhaps, authors brought in the concept of velocity because of the definition of critical length (L_c) which is defined as “the maximum distance a charge carrier can travel at an average velocity within a single hopping time”. According to their definition, L_c depends on the velocity, but it correlates with film thickness due to the field dependent ($v = \mu E$). In that case, the critical length will depend on thickness, and it could change with thickness. Did authors examine different active layer thicknesses? What thickness was used in current analysis?
4. To mimic the variation of electric field in the device due to variation in thickness (100 nm to 300 nm), the authors select the voltage range of 1 to 2V during the conductivity measurement. In that case, what is the thickness of the film used during the test (I must repeat the same question)? Why not vary the thickness directly to better reflect real device conditions? Clarify the test structure (e.g., metal electrode configuration), film thickness, and measurement geometry for reproducibility.
5. In Fig 2b, the hopping frequency appears to increase from $\sim 3.5 \times 10^5$ rad/s (1 V) to $\sim 1.2 \times 10^6$ rad/s (2 V) (judged from the shaded purple and pink area), suggesting a Δ of $\sim 8.5 \times 10^5$ rad/s for PM6:BTP-eC9—contradicting the reported value of 2.54. If this interpretation is incorrect, the presentation should be revised to avoid confusion. Additionally, Δ should retain units (rad/s) unless normalized; clarify its unitless status.
6. In Fig 2c, the blue square dots likely represent Δ values, but this should be explicitly stated in the caption. Are they linked to the top X-axis? What do the red circles signify (bottom X-axis)? Explain the connecting lines between dots and the purple/pink shading.
7. Please provide the source of Equation (1). If derived by the authors, include the derivation in the SI.
8. What does n in equation (2) refer to?
9. In line 175 of SI, the zero-field mobility should be μ_0 , instead of μ . Please confirm.

10. In equation 12 in SI, μ should likely denote field-dependent mobility (not zero-field mobility). If zero-field mobility is intended, use μ_0 . Also, β is defined inconsistently (field-dependant of mobility vs the Poole-Frenkel slope). Please standardize the terminology.

11. The electric field intensity is important to use the Poole-Frenkel model and associated calculations of L_c but it is not known how the electric field was determined. What is the electric field in the devices for the J-V curve in Fig 2d?

12. Authors observed that defect density and Urbach energy are reduced. I think reduced defect density and Urbach energy could help to increase critical length. The detailed explanation on the connection between these parameters will be interesting and make the manuscript more coherent. Currently, there is lack of connection. The analysis of the defect density, Urbach energy, and transient absorption analysis seems to be a separate topic without linking to the critical length analysis.

13. The acceptor domain size became smaller in 300 nm-thick film compared to 100 nm-thick film for many systems, especially D18:L8-BO. This clearly shows that film thickness has strong influence on morphology, in turn, domain size. I believe that changes in morphology and domain size would affect the critical length, and hence it is important to study the thickness effect on the critical length.

14. A major weakness of the manuscript is lack of focus and coherence. For example, the manuscript inconsistently switches between PM6 (critical length analysis) and D18 (device analysis). Without evidence that PM6-based trends apply to D18 systems, comparisons are unclear. Maintain consistency in material systems for valid conclusions.

15. According to the claim, the critical length is an important parameter for device performance. In that case, how is the device performance of D18:BTP-eC9 which contains higher content of BTP-eC9 with longer critical length compared to D18:L8-BO:BTP-eC9?

Reviewer #2

(Remarks to the Author)

Meng and coauthors demonstrate a new model for screening photovoltaic materials for thick-film Organic Solar Cells (OSCs), introducing "critical length" as a key screening metric. This approach addresses a long-standing limitation of relying solely on carrier mobility. The authors claim that the critical length is a key factor for a range of acceptor materials, including both fullerene and non-fullerene materials, and identify BTP-eC9 and L8-BO as promising candidates. Subsequently, the authors fabricate a series of OSCs based on BTP-eC9, achieving a high Power Conversion Efficiency (PCE) exceeding 19% in thick-film OSCs. The concept of critical length provides a novel perspective that may help guide the design of thick-film devices. The critical length introduced in this article incorporates various steady-state properties, including mobility and AC conductivity. Performance correlations for thick-film devices are effectively established through the testing of these parameters. This novel experimental approach, linking microstructural morphology to overall device performance through classical characterization techniques, holds significant importance. However, the current presentation lacks details for this central part of the work. It is recommended that the authors expand and clearly discuss this section, enabling a broader audience of researchers in the field to become familiar with the method. This will enhance both the accessibility and the appeal of the article to a wider readership. Overall, this work is nicely presented, and thus, this work could be published in Nature Communications following the revisions commented below:

(1) Figure 1c represents a correlation between the critical length and the average carrier velocity, but there is no term related to the average velocity in Equation 1, so Equation 1 needs further explanation.

(2) From Equation 1, the authors suggest a low hopping frequency is needed to obtain a large critical length, which contradicts the traditional expectation that a low hopping frequency means lower mobility. Clarifying this point will help readers better understand the physical implications of the model.

(3) The wavelength ranges for extracting localized excitons, hole transfer, and other transient spectral signals must be clearly specified in transient spectral characterization.

(4) It is necessary to clarify which recombination mechanism dominates in the D18:N3 and D18:L8-BO systems, respectively, regarding carrier recombination mechanisms.

(5) Several inconsistencies are noted in the formatting of the references, particularly in the use of capitalization and subscripts/superscripts in titles (e.g., Reference 22 in the supplementary file). It is recommended that the references be carefully reviewed and formatted in accordance with the journal's guidelines prior to resubmission.

(6) The improvement in device performance is attributed to increases in short-circuit current density (JSC) and fill factor (FF). What are the primary factors responsible for the improvement in FF? Please provide a systematic explanation.

(7) The morphological features of the blend systems have been characterized. It is essential to clarify which morphological features contribute to the improvement of JSC and FF mentioned in the article.

(8) A few minor corrections are needed. For example, when referring to multiple figures, plural forms should be used (e.g., "Figures 1 and 2"). In addition, the x-axis labels in Figure 7 appear too close to the axis and may require better spacing for readability.

Reviewer #3

(Remarks to the Author)

In this study, the authors propose the critical length as a predictor for thick-film device performance and demonstrate its broad applicability through a detailed investigation of various acceptor materials. This method represents a meaningful advancement toward the scalable fabrication of high-performance organic photovoltaics (OPVs). Importantly, the study establishes a quantitative relationship between critical length and film morphology (particularly with domain size), offering a practical and visual guideline for optimizing thick-film device performance. Using such a method, the authors achieve a record-high power conversion efficiency (PCE) in thick-film devices. The charge transport modeling with device fabrication and morphological results in a coherent and compelling data description. Therefore, I am pleased to recommend an acceptance decision for this manuscript to be published in Nature Communications after minor revisions. Authors need to fully address the required issues listed below:

1. The manuscript attempts to introduce the physical basis of the critical length concept; however, it remains to be clarified whether this concept applies to more complex systems involving organic donor-acceptor blends. Clarification is needed on whether the model addresses static (energetic) or dynamic (vibrational) disorder.
2. The discussion on the critical length is informative. However, what is the definition or extraction method used to determine the hopping frequency? Additionally, are there alternative models that could be considered for determining the hopping frequency?
3. The article presents extensive characterization data for both the D18:L8-BO and D18:N3 systems. What is the common impact of high critical length acceptor materials on device performance in both systems?
4. The critical length presented in the manuscript is determined by three parameters, each reflecting certain characteristics of charge carriers. To enhance the physical insight and clarity of the model, it would be beneficial to provide more specific physical interpretations of these parameters in the context of charge transport mechanisms.
5. Transient absorption spectroscopy analyzes exciton and hole transfer dynamics and identifies the optimal BTP-eC9 ratio in both systems. Is this ratio consistent with the actual ratio used during device fabrication? The manuscript should clearly state the specific amount of BTP-eC9 added in the device preparation process.

Version 1:

Reviewer comments:

Reviewer #1

(Remarks to the Author)

The authors have thoroughly addressed all the comments and concern from the reviewers. So, I am pleased to accept manuscript in its current form.

Reviewer #2

(Remarks to the Author)

Revision satisfied.

Reviewer #3

(Remarks to the Author)

I have no more comments on this submission. I recommend acceptance of the manuscript.

Reviewer #1 (Remarks to the Author):

This manuscript investigates the critical length in organic photovoltaic (OPV) devices, analyzing its dependence on three key parameters: mobility, field-dependent mobility, and hopping frequency. The experimental results reveal that higher mobility does not always correlate with longer critical length. Instead, the critical length is determined by the interplay of these three factors. Among the studied systems, PM6 paired with BTP-eC9 or L8-BO exhibits a longer critical length compared to other acceptors (ITIC, Y5, and PCBM). Notably, incorporating BTP-eC9 into the D18:L8-BO system further extended the critical length, enabling a power conversion efficiency (PCE) of 19% in a 300 nm-thick active layer device.

The study underscores the significance of critical length in OPVs, particularly for thick-film devices relevant to commercialization. However, the manuscript's current presentation lacks clarity in some areas, and certain analyses are insufficient to fully support the conclusions. Due to these issues, a comprehensive assessment is challenging at this stage. I recommend revising the manuscript to improve data presentation and analytical rigor before further evaluation. A second review may be necessary once additional details are provided.

1. *In Figure 1c, which material system does the data represent? Is this trend consistent across other systems studied?*

Response: In **Figure 1c**, we illustrated a general relationship among the average velocity, hopping frequency, and critical length, which is independent with the selection of donor:acceptor bulk-heterojunction systems. We elaborated on the relationship among these three parameters in our response to the second comment of Reviewer 1#. Using the measured zero-field mobility (μ_0) of the organic systems and field-dependent of mobility (β), combined with values from the literature,¹⁻³ we selected parameter ranges to generate **Figure 1c**. Based on the critical length model proposed by Papathanassiou et al.,⁴ this approach captures the relationship among the characteristic average velocity, critical length, and hopping frequency in organic systems. The 3D

plot in **Figure 1c** enables analysis of how the average velocity and hopping frequency influence the critical length, providing a basis for screening acceptor materials with large critical lengths to fabricate high-efficiency thick-film organic solar cells.

2. *The manuscript mentions using SCLC method to determine average carrier velocity. However, equation (1) and Poole-Frenkel model do not explicitly involve carrier velocity. Using zero field mobility and electric field intensity, one can determine the critical length without knowing the average carrier velocity. So, I don't get the point why average carrier velocity was brought in this analysis. Furthermore, according to Ref 37 and my knowledge, SCLC method can provide the mobility only and it is unclear how the carrier velocity was determined. Perhaps, authors used the relationship between carrier drift velocity and mobility ($v = \mu E$), but it should be clearly stated. Additionally, since velocity is field-dependent, it may vary with film thickness—was this accounted for?*

Response: We sincerely thank the reviewer for the valuable comments, which helped us further improve the model description and enhance the overall clarity of the manuscript. We condensed the reviewer comment 2# into three sub-questions: (i) the necessity of the average carrier velocity, (ii) determining the average velocity, and (iii) the thickness dependence of carrier velocity.

First, we introduced the concept of critical length to screen acceptors for efficient thick-film organic solar cells (OSCs). Since its initial definition is the maximum distance a charge carrier can travel at an average velocity within a single hopping time, determining the average velocity is essential. However, applying $v = \mu E$ to the calculation of the average carrier velocity in OSCs is rather challenging due to highly nonuniform and time-dependent electric field across the active layer.

The expression $v = \mu E$, where μ is the field-dependent mobility and E is the electric field strength, is valid under the assumption of a spatially uniform and

temporally steady electric field. However, such conditions are generally not realized in practical OSCs, and therefore this expression does not accurately describe charge carrier transport in these devices. In OSCs, the built-in electric field arises from the work function difference between electrodes and is strongly affected by factors such as space-charge accumulation, dielectric heterogeneity, and local polarization. These effects lead to a highly nonuniform and time-dependent electric field across the active layer.^{5,6} As a result, the electric field varies significantly with position.^{7,8} Therefore, under such nonuniform field conditions, it is challenging to apply $v = \mu E$ for analyzing the average carrier velocity in OSCs.

Second, charge carrier transport in the active layer is governed, on one hand, by the intrinsic zero-field mobility (μ_0), which reflects the capability of charge carriers to move under vanishing fields.⁹ On the other hand, the overall energy landscape fluctuations induced by defect states and static charges - through Coulombic potential fluctuations - play a crucial role in determining the charge carrier migration and diffusion pathways. β reflects the degree to which the external electric field lowers the energy barrier for a carrier to escape from an isolated trap and is closely related to the dielectric properties of the material. A smaller β implies a weaker field dependence, indicating that the mobility remains relatively high even under low-field conditions, which is favorable for long-range charge carrier transport. In contrast, a larger β signifies that the mobility drops significantly in weak fields, hindering long-range transport. From a macroscopic perspective, β reflects the degree of energetic landscape fluctuations. A large β reflects more pronounced energetic disorder and stronger field dependence of mobility. This interpretation is consistent with the revised PF framework where β characterizes how the underlying energy disorder governs field-activated transport.¹⁰

It should be noted that β simultaneously characterizes two aspects of charge carrier transport in active layers: (i) the escape probability of charge carriers from individual defect states; and (ii) the impact of overall energetic landscape fluctuations on charge carrier transport pathways. By combining the intrinsic mobility (μ_0) and the field-

dependent of mobility (β), we construct a characteristic velocity ($v = \frac{\mu_0}{\beta^2}$) within two transport aspects mentioned above. Dimensional analysis further confirms that this velocity is consistent with the average velocity of charge carriers, thus providing a realistic description of charge carrier transport in organic photovoltaic devices under non-steady-state fields and complex energetic landscapes.

Additionally, in AC conductivity measurements, the applied voltage is sinusoidal, alternating between forward and reverse field directions. These opposing field components respectively promote and hinder carrier extraction at the cathode. To account for this in the critical length estimation, we consider only the forward half-cycle of the sinusoidal field, and therefore adopt half the period ($t=1/2f=1/2\omega$) in the denominator. As a result, the full expression for the critical length includes the factor $2\omega_H$ in the denominator.

In summary, the critical length of charge carriers in organic solar cells is calculated using Equation (1) in the manuscript. Following the reviewer's suggestion, we have added the original definition of the critical length to the revised ESI, along with the above detailed explanation to clarify the basis of our approach.

Thirdly, regarding the thickness dependence of charge carrier transport, we have added systematic explanations and supplementary experiments in the revised manuscript and ESI (see responses to reviewer comments 3# & 13#). The detailed results and analysis can be found in our response to reviewer comments 3#, as well as in the newly added **Figures R2-R5** and **Tables R1** and **R2**.

3. *Perhaps, authors brought in the concept of velocity because of the definition of critical length (L_c) which is defined as “the maximum distance a charge carrier can travel at an average velocity within a single hopping time”. According to their definition, L_c depends on the velocity, but it correlates with film thickness due to the*

field dependent ($v = \mu E$). In that case, the critical length will depend on thickness, and it could change with thickness. Did authors examine different active layer thicknesses? What thickness was used in current analysis?

Response: We appreciate the reviewer for the valuable comment. Regarding the effect of thickness, we have already applied normalization during the measurement process. First, in our conductivity measurements, we normalized the results with respect to film thickness using the intrinsic conductivity equation: $\sigma = \frac{L}{R \times A}$, where L is the active layer thickness, A is the effective area, and R is the resistance. Similarly, when calculating the zero-field mobility and the β parameter using the PF model, we also normalized by device thickness (i.e., L in the equation corresponds to the active layer thickness). Therefore, in the calculation of the critical length, the effect of film thickness was already normalized.

Furthermore, the analysis of charge carrier transport experiments in OSCs, the thickness of active layers must exceed a certain value (usually ~ 200 nm). Otherwise, interfacial effects such as Ohmic contact and interface recombination become more dominant than the bulk transport effects, resulting in inaccurate measurements.^{11,12} Non-Ohmic contacts can form injection barriers at the interface of the active layer, hindering the entry of charge carriers into the active layer. In thinner devices, the injection barrier becomes more pronounced, and the accumulation of charges at the interface generates an electric field that distorts the current–voltage characteristics. Increasing the thickness of the active layer (above 100 nm) facilitates the formation of near-Ohmic contacts (with injection barriers below 0.4 eV), as shown in **Figure R1**.

Figure R1. Extracted mobility using analytical equations when the injection barriers are increased (0.1, 0.2 and 0.3 eV); (a) extraction of mobility when fitting with the MG law at the slope maximum, and (b) extraction of the mobility when fitting with the moving-electrode equation. The values are normalized with respect to the input value for the mobility.¹³

Based on the reviewer's suggestions, we have supplemented the analysis of charge carrier transport under different thicknesses for the PM6:BTP-eC9 and PM6:ITIC systems, as shown in **Figures R2-R5** and **Tables R1** and **R2**. When the active layer thickness is 100 nm, due to the dominant interface effect, the zero-field mobility μ_0 , field-dependent of mobility β , and hopping frequency ω_H all differ significantly from those at a thickness of 300 nm. When the active layer thickness of both systems is around 100 nm, the carrier injection barrier becomes relatively high, leading to distortions in the current density-voltage (J - V) curves. This deviation from the ideal model fit results in inaccurate determination of both the zero-field μ_0 and its electric field dependence β . Once the active layer thickness reaches a certain value (≥ 200 nm), the charge carrier transport analysis, including μ_0 , β , and ω_H , tends to be stabilized. This indicates that our analysis of charge carrier transport is based on the bulk effect of charge carrier transport. Therefore, using an active layer thickness of 300 nm is reasonable.

Importantly, active layer of 300 nm is not only appropriate for transport measurements but also of practical relevance. Such thickness enables effective light absorption,

leading to improved short-circuit current density.¹⁴ The thick-film structure is less sensitive to minor thickness fluctuations during fabrication, which is advantageous for ensuring process stability in large-scale manufacturing techniques such as roll-to-roll printing, thereby enhancing the feasibility and yield of devices in industrial production.¹⁵ Considering both the practical relevance to large-scale device deployment and the requirements of charge transport analysis, we selected an active layer thickness of 300 nm.

Finally, the main objective of our study is to use critical length screening to guide the fabrication of high-efficiency thick-film organic solar cells. The thick-film devices we fabricated have active layer thicknesses of approximately 300 nm, which is consistent with the thickness of electron-only devices used in our earlier critical length analysis. Therefore, the charge carrier transport analysis and device fabrication are well aligned in terms of active layer thickness. In the section **Electron-only Devices Fabrication and Characterization** on page 2 of the revised ESI, the thickness of the active layer has been specified.

Figure R2. Variation of zero-field mobility (μ_0), field-dependent of mobility (β), hopping frequency (ω_H), and critical length (L_C) with active layer thickness in the PM6:BTP-eC9 system.

Figure R3. Variation of zero-field mobility (μ_0), field-dependent of mobility (β), hopping frequency (ω_H), and critical length (L_c) with active layer thickness in the PM6:ITIC system.

Figure R4. Extraction of critical length for PM6:BTP-eC9 devices with different active layer thicknesses. (a) Conductivity–frequency plot for a 100 nm active layer. (b) Conductivity–frequency plot for a 200 nm active layer. (c) Conductivity–frequency plot for a 400 nm active layer. (d) Current density–voltage (J – V) characteristics.

Figure R5. Extraction of critical length for PM6:ITIC devices with different active layer thicknesses. (a) Conductivity–frequency plot for a 100 nm active layer. (b) Conductivity–frequency plot for a 200 nm active layer. (c) Conductivity–frequency

plot for a 400 nm active layer. (d) Current density–voltage (J – V) characteristics.

Table R1. Zero-field mobility, field-dependent of mobility and critical length of PM6:BTP-eC9 system.

Thickness	μ_0 (cm ² V ⁻¹ s ⁻¹)	β (cm ^{1/2} V ^{-1/2})	ω_H with 1 V voltage (rad/s)	ω_H with 2 V voltage (rad/s)	L_C (nm)
100	2.19×10^{-4}	1.33×10^{-3}	3.00×10^5	3.61×10^5	
200	1.05×10^{-4}	6.23×10^{-3}	3.90×10^5	1.93×10^6	34.6
400	9.42×10^{-5}	5.87×10^{-3}	3.98×10^5	2.24×10^6	34.5

Table R2. Zero-field mobility, field-dependent of mobility and critical length of PM6:ITIC system.

Thickness	μ_0 (cm ² V ⁻¹ s ⁻¹)	β (cm ^{1/2} V ^{-1/2})	ω_H with 1 V voltage (rad/s)	ω_H with 2 V voltage (rad/s)	L_C (nm)
100	1.06×10^{-6}	5.41×10^{-3}	1.11×10^5	6.46×10^5	1.63
200	8.87×10^{-6}	5.88×10^{-3}	1.15×10^5	8.40×10^6	11.19
400	6.69×10^{-6}	8.88×10^{-3}	2.54×10^4	3.77×10^5	16.66

4. *To mimic the variation of electric field in the device due to variation in thickness (100 nm to 300 nm), the authors select the voltage range of 1 to 2V during the conductivity measurement. In that case, what is the thickness of the film used during the test (I must repeat the same question)? Why not vary the thickness directly to better reflect real device conditions? Clarify the test structure (e.g., metal electrode configuration), film thickness, and measurement geometry for reproducibility.*

Response: We appreciate the reviewer for the valuable comment. Our study aims to use the critical length screening to guide the fabrication of high-efficiency thick-film organic solar cells. The devices we fabricated have an active-layer thickness of about 300 nm, which is the similar thickness of active layers used in our earlier critical length

calculations.

Accurate analysis of charge carrier transport requires the active layer to be above a certain thickness. At around 100 nm, interfacial effects - such as Ohmic contact and interface recombination^{11,12} - dominate, giving zero-field mobility, field-dependent of mobility, and hopping-frequency values that differ greatly from those at 300 nm, as shown in the data for reviewer comment 3#. For this reason, we used a 300 nm active layer in the transport experiments. The effect of electric field variation on charge carrier hopping frequency was then examined by applying different voltages to devices of this thickness.

The full device structure and active layer thickness is described in the **Electron-only Devices Fabrication and Characterization** section of the revised ESI.

5. *In Fig 2b, the hopping frequency appears to increase from $\sim 3.5 \times 10^5$ rad/s (1 V) to $\sim 1.2 \times 10^6$ rad/s (2 V) (judged from the shaded purple and pink area), suggesting a Δ of $\sim 8.5 \times 10^5$ rad/s for PM6:BTP-eC9—contradicting the reported value of 2.54. If this interpretation is incorrect, the presentation should be revised to avoid confusion. Additionally, Δ should retain units (rad/s) unless normalized; clarify its unitless status.*

Response: We appreciate the reviewer for the valuable comment. The parameter Δ is defined as the relative change in hopping frequency when the applied voltage increases from 1 V to 2 V. Specifically, Δ is calculated as the difference in hopping frequency between 2 V and 1 V, normalized by the hopping frequency at 1 V. Therefore, Δ is a dimensionless and normalized parameter that reflects how much the hopping frequency changes relative to its value at 1 V.

We have added detailed description for Δ on page 10 of the revised ESI. On page 8 of the revised manuscript, we have added the following description: “The detailed

description of Δ are given in the ESI, where Δ is a dimensionless parameter.”

6. *In Fig 2c, the blue square dots likely represent Δ values, but this should be explicitly stated in the caption. Are they linked to the top X-axis? What do the red circles signify (bottom X-axis)? Explain the connecting lines between dots and the purple/pink shading.*

Response: We appreciate the reviewer for the valuable comment. In **Figure 2c**, we have integrated both the hopping frequency and its relative change from 1 V to 2 V into a single comprehensive plot. The red circles represent the absolute hopping frequencies of different systems, while the blue dots indicate the corresponding relative change, Δ . The gray connecting lines link the two parameters belonging to the same system.

The blue-shaded region indicates lower values of both hopping frequency and relative change, whereas the red-shaded region corresponds to higher values of these two parameters. The upper X-axis shows the relative change in hopping frequency (Δ), while the bottom X-axis shows the absolute hopping frequency.

On page 8 of the revised manuscript, we have added a detailed explanation in the figure caption to improve the clarity.

7. *Please provide the source of Equation (1). If derived by the authors, include the derivation in the SI.*

Response: We appreciate the reviewer for the valuable comment. The concept of critical length was originally proposed by Papathanassiou et al. to describe the distribution of conductive pathways in the universal AC conductivity response model.⁴ It was defined as the ratio of the average carrier velocity to the hopping frequency (the inverse of the hopping time).

Charge carrier transport in the active layer is governed, on one hand, by the intrinsic zero-field mobility (μ_0), which reflects the fundamental capability of carriers to move under vanishing fields. On the other hand, the overall energy landscape fluctuations induced by defect states and static charges- through Coulombic potential fluctuations - play a crucial role in determining the carrier's migration and diffusion pathways. β reflects the degree to which the external electric field lowers the energy barrier for a carrier to escape from an isolated trap and is closely related to the dielectric properties of the material. From a broader perspective, the value of β serves as a proxy for the smoothness or ruggedness of the energy landscape. Therefore, we combined the intrinsic mobility (μ_0) and field-dependent of mobility (β) to construct a meaningful characteristic velocity.

Additionally, in AC conductivity measurements, the applied voltage is sinusoidal, alternating between forward and reverse field directions. These opposing field components respectively promote and hinder carrier extraction at the cathode. To account for this in the critical length estimation, we consider only the forward half-cycle of the sinusoidal field, and therefore adopt half the period ($t=1/2f=1/2\omega$) in the denominator. As a result, the full expression for the critical length includes the factor $2\omega_H$ in the denominator.

In summary, the critical length of charge carriers in organic solar cells is calculated using Equation (1) in the manuscript. Following the reviewer's suggestion, we have added the relevant explanation for Equation 1 on pages 8 and 9 of the revised ESI.

8. *What does n in equation (2) refer to?*

Response: n represents the frequency exponent in the high-frequency region of the conductivity. To clarify this, we have added an explicit explanation of n in the annotation of Equation 2 on page 6 of the revised manuscript.

9. *In line 175 of SI, the zero-field mobility should be μ_0 , instead of μ . Please confirm.*

Response: We appreciate the reviewer for the valuable comment. We have updated both the revised manuscript and ESI to consistently use μ_0 to represent the zero-field mobility.

10. *In equation 12 in SI, μ should likely denote field-dependent mobility (not zero-field mobility). If zero-field mobility is intended, use μ_0 . Also, β is defined inconsistently (field-dependant of mobility vs the Poole-Frenkel slope). Please standardize the terminology.*

Response: In the revised manuscript, we have consistently used the symbol μ_0 to denote the zero-field mobility. Meanwhile, for consistency, we have revised both the revised manuscript and the ESI to uniformly describe β as the field-dependent of mobility. We sincerely thank the reviewer for the careful and constructive suggestions, which have greatly helped improve the rigor and precision of our manuscript.

11. *The electric field intensity is important to use the Poole-Frenkel model and associated calculations of L_c but it is not known how the electric field was determined. What is the electric field in the devices for the J - V curve in Fig 2d?*

Response: The J - V curves shown in **Figure 2d** were obtained using the space-charge-limited current (SCLC) method. A continuous DC voltage was applied to the device, and the resulting current was recorded. From the voltage-current relationship, the zero-field mobility and field-dependent of mobility were extracted. In the SCLC method, the applied voltage provides the driving force for charge carrier transport. Variations in defect distribution, space-charge distribution, and active-layer morphology can lead to different current magnitudes and trends in current-voltage response. Thus, the charge carrier transport behavior within the active layer is directly reflected in the current response, with the voltage serving as the driving force.

12. *Authors observed that defect density and Urbach energy are reduced. I think reduced defect density and Urbach energy could help to increase critical length. The detailed explanation on the correction between these parameters will be interesting and make the manuscript more coherent. Currently, there is lack of connection. The analysis of the defect density, Urbach energy, and transient absorption analysis seems to be a separate topic without linking to the critical length analysis.*

Response: We thank the reviewer for the valuable comment and fully agree that further clarifying the relationship between defect density, Urbach energy, and critical length will enhance the logical flow and completeness of the manuscript.

A reduction in defect density means fewer trap states, which prolongs carrier lifetime and reduces the recombination probability, allowing carriers to travel longer distances before being trapped. This naturally contributes to an increase in the critical length. We further investigated the recombination processes and carrier lifetimes within the devices. The suppressed recombination and prolonged carrier lifetimes after incorporating BTP-eC9 corroborate our conclusions.

Meanwhile, a lower Urbach energy (E_U) indicates reduced energetic disorder in the material, leading to a narrower tail in the density of states and a flatter transport energy landscape.¹⁶ This is particularly important for thick-film devices, where charge carriers must traverse longer distances to reach the electrodes. A reduced E_U decreases the density of low-energy tail states, thereby lowering the likelihood of carrier trapping during transport. Consequently, the critical length is extended, facilitating more efficient charge collection and preserving device performance even at larger active layer thicknesses.

These two effects work synergistically to enlarge the spatial range over which

photogenerated carriers can propagate before recombination, which is consistent with the definition and extraction of critical length. We have incorporated a detailed discussion on pages 17 and 19 of the revised manuscript.

13. The acceptor domain size became smaller in 300 nm-thick film compared to 100 nm-thick film for many systems, especially D18:L8-BO. This clearly shows that film thickness has strong influence on morphology, in turn, domain size. I believe that changes in morphology and domain size would affect the critical length, and hence it is important to study the thickness effect on the critical length.

Response: We appreciate the reviewer for the valuable comment. Our study aims to use critical length screening to guide the fabrication of high-efficiency thick-film OSCs. The thick-film devices we fabricated have active-layer thicknesses of approximately 300 nm, which is the same thickness used in our earlier critical length calculations. A 300 nm active layer is not only suitable for transport measurements, but also of practical importance: it enables efficient solar light absorption, leading to higher short-circuit current density,¹⁴ and meets the requirements for roll-to-roll large-scale production.¹⁵

We agree with the reviewer that active-layer morphology changes with thickness. However, as we noted in our response to reviewer comment 3#, due to strong interfacial effects at 100 nm, it is difficult to obtain reliable transport measurements that directly correlate the morphology of 100 nm-thick films. Moreover, in organic solar cell fabrication, active-layer morphology is determined not only by thickness but also by multiple factors such as donor-acceptor ratio,¹⁷ spin-coating speed,¹⁸ solution concentration,¹⁹ and post-treatment conditions.²⁰ Even at the same thickness, variations in these fabrication parameters can result in different fiber-network structures and domain sizes. We have verified this experimentally:

1. preparing films of different thicknesses using different concentrations but the same spin speed;

2. preparing films of different thicknesses using the same concentration but different spin speeds.

AFM and GISAXS measurements on films of the same thickness but prepared under different conditions revealed differences in fiber-network structure and domain size. Therefore, morphology cannot be reliably inferred from thickness alone. The results are shown in **Figures R6–R8** and **Table R3**. (The AFM and GISAXS data of D18:L8-BO system prepared from a 4.5 mg/mL solution at a spin speed of 3000 rpm (corresponding to a thickness of ~100 nm) and from a 9 mg/mL solution at 3000 rpm (corresponding to a thickness of ~300 nm) are provided in the main text. Detailed results are shown in Figure S27 and Table S6.)

For the fabrication of 100-nm-thick films, a solution concentration of 4.5 mg/mL and a spin speed of 3000 rpm/s were used, while for the 300-nm-thick active layers, the concentration was increased to 9 mg/mL with the same spin speed of 3000 rpm/s. *We have ensured that transport experiments, thick-film device fabrication, and morphology characterizations are performed under identical fabrication conditions and with the same thickness, so that transport and morphology analyses remain consistent and jointly inform the design of thick-film devices.*

Figure R6. AFM and GISAXS images of 100-nm D18:L8-BO active layers prepared at different solution concentrations and spin-coating speeds. (a) AFM image of film prepared from a 4.5 mg/mL solution and spin-coated at 3000 rpm/s. (b) AFM image of film prepared from a 6.5 mg/mL solution and spin-coated at 4000 rpm/s. (c) AFM image of film prepared from a 9 mg/mL solution and spin-coated at 6000 rpm/s. (d) GISAXS image of film prepared from a 4.5 mg/mL solution and spin-coated at 3000 rpm/s. (e) GISAXS image of film prepared from a 6.5 mg/mL solution and spin-coated at 4000 rpm/s. (f) GISAXS image of film prepared from a 9 mg/mL solution and spin-coated at 6000 rpm/s.

Figure R7. AFM and GISAXS images of 200-nm D18:L8-BO active layers prepared at different solution concentrations and spin-coating speeds. (a) AFM image of film prepared from a 6.5 mg/mL solution and spin-coated at 3000 rpm/s. (b) AFM image of film prepared from a 9 mg/mL solution and spin-coated at 4000 rpm/s. (c) AFM image of film prepared from a 11 mg/mL solution and spin-coated at 6000 rpm/s. (d) GISAXS image of film prepared from a 6.5 mg/mL solution and spin-coated at 3000 rpm/s. (e) GISAXS image of film prepared from a 9 mg/mL solution and spin-coated at 4000 rpm/s. (f) GISAXS image of film prepared from a 11 mg/mL solution and spin-coated at 6000 rpm/s.

Figure R8. AFM and GISAXS images of 300-nm D18:L8-BO active layers prepared at different solution concentrations and spin-coating speeds. (a) AFM image of film prepared from a 9 mg/mL solution and spin-coated at 3000 rpm/s. (b) AFM image of film prepared from a 11 mg/mL solution and spin-coated at 4000 rpm/s. (c) GISAXS image of film prepared from a 9 mg/mL solution and spin-coated at 3000 rpm/s. (d) GISAXS image of film prepared from a 11 mg/mL solution and spin-coated at 4000 rpm/s.

Figure R9. Fractal fitting of one-dimensional lines extracted from 2D GISAXS images.

(a) Extracted line-cut profiles of the D18:L8-BO system prepared from a 6.5 mg/mL solution and spin-coated at 4000 rpm/s. (b) Extracted line-cut profiles of the D18:L8-BO system prepared from a 9 mg/mL solution and spin-coated at 6000 rpm/s. (c) Extracted line-cut profiles of the D18:L8-BO system prepared from a 6.5 mg/mL solution and spin-coated at 3000 rpm/s. (d) Extracted line-cut profiles of the D18:L8-BO system prepared from a 9 mg/mL solution and spin-coated at 4000 rpm/s. (e) Extracted line-cut profiles of the D18:L8-BO system prepared from a 11 mg/mL solution and spin-coated at 6000 rpm/s. (f) Extracted line-cut profiles of the D18:L8-BO system prepared from a 11 mg/mL solution and spin-coated at 4000 rpm/s.

Table R3. The acceptor domain size of the D18:L8-BO system.

	D	η (nm)	R_g (nm)
4.5 mg/ml+3000 rpm/s (100 nm)	2.92	17.87	42.73
6.5 mg/ml+4000 rpm/s (100 nm)	2.89	11.15	26.44
9 mg/ml+6000 rpm/s (100 nm)	2.65	10.13	22.28
6.5 mg/ml+3000 rpm/s (200 nm)	2.95	10.60	25.59

9 mg/ml+4000 rpm/s (200 nm)	2.98	10.50	25.57
11 mg/ml+6000 rpm/s (200 nm)	2.84	13.33	31.12
9 mg/ml+3000 rpm/s (300 nm)	2.96	6.87	16.62
11 mg/ml+4000 rpm/s (300 nm)	2.99	9.58	23.39

For the D18:L8-BO system, we also examined charge carrier transport in active layers of different thicknesses, with the results summarized in **Figures R10–R12** and **Table R4**. For the 100-nm active layer, both the current density-voltage (J - V) curves and the conductivity-frequency profiles show noticeable deviations. As a result, fitting the mobility and its field dependence using the Poole–Frenkel model may not fully capture the actual charge carrier dynamics, and applying the Almond–West formalism to the conductivity curves provides hopping frequencies that are somewhat less accurate. Consequently, the critical length extracted for the 100 nm-thick active layer differs more markedly from those obtained for layers with 200, 300, and 400 nm thickness. By contrast, for active layers of 200 nm or thicker, the zero-field mobility, field-dependent of the mobility, hopping frequency, and critical length are relatively consistent. *These observations suggest that using a 300 nm-thick active layer for transport analysis is likely to provide a more representative description of the intrinsic carrier behavior in the bulk, with reduced influence from interfacial effects such as Ohmic contacts.*

Figure R10. Variation of zero-field mobility (μ_0), field-dependent of mobility (β), hopping frequency (ω_H), and critical length (L_C) with active layer thickness in the D18:L8-BO system.

Figure R11. Conductivity–frequency plots of D18:L8-BO devices with different active layer thicknesses. (a) 100 nm. (b) 200 nm. (c) 300 nm. (d) 400 nm.

Figure R12. Current density–voltage (J – V) characteristics of D18:L8-BO devices with varying active layer thicknesses.

Table R4. Zero-field mobility, field-dependent of mobility and critical length of D18:L8-BO system.

Thickness	μ_0 (cm ² V ⁻¹ s ⁻¹)	β (cm ^{1/2} V ^{-1/2})	ω_H with 1 V voltage (rad/s)	ω_H with 2 V voltage (rad/s)	L_C (nm)
100	3.83×10^{-5}	2.87×10^{-3}	4.73×10^5	8.96×10^5	49.17
200	4.90×10^{-5}	4.90×10^{-3}	2.16×10^5	5.45×10^5	37.59
300	7.50×10^{-5}	5.13×10^{-3}	3.67×10^5	8.44×10^5	38.97
400	7.28×10^{-5}	5.73×10^{-3}	3.03×10^5	1.13×10^6	36.65

14. A major weakness of the manuscript is lack of focus and coherence. For example, the manuscript inconsistently switches between PM6 (critical length analysis) and D18 (device analysis). Without evidence that PM6-based trends apply to D18 systems, comparisons are unclear. Maintain consistency in material systems for valid conclusions.

Response: We sincerely appreciate the reviewer for the valuable comment. We fully agree with the reviewer that maintaining a consistent donor is indeed beneficial for obtaining more coherent and meaningful data insights.

In our initial experiments on the charge carrier transport, we conducted a systematic analysis of a series of acceptors using the same polymer donor, PM6. As a commonly used donor in organic solar cells, PM6 exhibits moderate energy levels and desirable compatibility with various acceptors.²¹⁻²³ In later studies, we employed the polymer donor D18, whose more pronounced fibrillar network structure leads to higher device performance in OSC fabrication.²⁴ Accordingly, the acceptors screened in the initial PM6-based experiments were incorporated into D18-based systems to achieve high-efficiency thick-film OSCs.

In this revision, we have supplemented our study with charge carrier transport

experiments for different acceptors using D18 as the donor. The results are shown in **Figures S30–S31** and **Table S19**. The critical lengths of the three acceptors with D18 as the donor are comparable to those with PM6, and they exhibit the same relative trend. The critical length in BTP-eC9 exceeds that observed in L8-BO, and both are greater than the critical length measured in IT-4F. On page 5 of the revised manuscript, we have added relevant discussion.

Figure S30. Conductivity–frequency plots of D18-based devices. (a) D18:BTP-eC9. (b) D18:IT-4F. (c) D18:L8-BO.

Figure S31. Current density–voltage (J – V) characteristics of D18-based devices.

Table S19. Zero-field mobility, field-dependent of mobility and critical length of D18-based systems.

	μ_0 (cm ² V ⁻¹ s ⁻¹)	β (cm ^{1/2} V ^{-1/2})	ω_H with 1 V voltage (rad/s)	ω_H with 2 V voltage (rad/s)	L_C (nm)
D18:BTP-eC9	1.50×10^{-4}	7.79×10^{-3}	3.07×10^5	3.34×10^5	40.19
D18:IT-4F	5.99×10^{-5}	5.79×10^{-3}	2.83×10^5	1.55×10^6	31.63
D18:L8-BO	7.50×10^{-5}	5.13×10^{-3}	3.67×10^5	8.44×10^5	38.97

15. According to the claim, the critical length is an important parameter for device performance. In that case, how is the device performance of D18:BTP-eC9 which contains higher content of BTP-eC9 with longer critical length compared to D18:L8-BO:BTP-eC9?

Response: We appreciate the reviewer for the valuable comment. The critical lengths of the D18:BTP-eC9, D18:L8-BO, and D18:L8-BO:BTP-eC9 systems have provided in **Tables S19** and **S1**, respectively, and are further summarized in **Table R5** for clarity. As shown, although the critical length of D18:BTP-eC9 is larger than that of D18:L8-BO, the ternary system D18:L8-BO:BTP-eC9 exhibits the longest value of 62.82 nm.

We have also included the device data for D18:BTP-eC9 organic solar cells, as presented in **Figure S28** and **Table S17**. The 300 nm-thick devices based on D18:BTP-eC9 show higher PCEs than those of D18:L8-BO, while the ternary D18:L8-BO:BTP-eC9 system, which possesses the largest critical length, delivers the highest PCE for the 300 nm-thick devices. These observations are consistent with our conclusion that a larger critical length is beneficial for enhancing the performance of thick-film devices.

In previous work, Zhan et al. investigated systems employing L8-BO and BTP-eC9 as dual acceptors.²⁵ The Flory–Huggins interaction parameter between L8-BO and BTP-eC9 was reported to be 0.69, indicating favorable miscibility. Moreover, BTP-eC9 and L8-BO were shown to form parallel phases: rather than interpenetrating each other's domains, they preferentially locate at the interfaces, thereby facilitating long-range

charge transport. This observation aligns well with our finding that the ternary D18:L8-BO:BTP-eC9 system exhibits the largest critical length.

To improve readability and strengthen the logical flow of the manuscript, we have added the corresponding explanation on page 10 of the revised manuscript.

Table R5. Zero-field mobility, field-dependent of mobility and critical length of D18-based systems.

	μ_0 (cm ² V ⁻¹ s ⁻¹)	β (cm ^{1/2} V ^{-1/2})	ω_H with 1 V voltage (rad/s)	L_C (nm)
D18:BTP-eC9	1.50×10^{-4}	7.79×10^{-3}	3.07×10^5	40.19
D18:L8-BO	5.99×10^{-5}	5.79×10^{-3}	2.83×10^5	31.63
D18:L8-BO:BTP-eC9	1.92×10^{-4}	6.20×10^{-3}	3.97×10^5	62.82

Table S17. Device performance of D18:BTP-eC9.

	Thickness (nm)	V_{OC} (V)	J_{SC}/J_{EQE} (mA/cm ²)	FF (%)	PCE (%)
D18:BTP-eC9	100	0.87	27.63/26.92	77.9	18.7
	300	0.86	28.10/27.67	69.3	16.7

Table S18. Device performance of 300 nm-thick D18:L8-BO:BTP-eC9 devices.

	Ratio	V_{OC} (V)	J_{SC} (mA/cm ²)	FF (%)	PCE (%)
D18:L8-BO:BTP-eC9	1:1.1:0.1	0.90	27.88	75.4	19.0
	1:1:0.2	0.90	27.79	74.0	18.5
	1:0.9:0.3	0.90	27.55	72.8	18.0

Figure S28. Current density–voltage (J – V) characteristics and external quantum efficiency (EQE) spectra of D18:BTP-eC9 devices. (a) J – V curve. (b) EQE spectrum.

Figure S29. Current density–voltage (J – V) curves of devices with varying BTP-eC9 ratios.

References:

1. Zhu, L. *et al.* Single-Junction Organic Solar Cells with Over 19% Efficiency Enabled by a Refined Double-Fibril Network Morphology. *Nat. Mater.* **21**, 656–663 (2022).
2. Fu, J. *et al.* 19.31% Binary Organic Solar Cell and Low Non-Radiative Recombination Enabled by Non-Monotonic Intermediate State Transition. *Nat.*

Commun. **14**, 1760 (2023).

3. Wu, J. *et al.* A Comparison of Charge Carrier Dynamics in Organic and Perovskite Solar Cells. *Adv. Mater.* **34**, 2101833 (2022).
4. Papathanassiou, A. N., Sakellis, I. & Grammatikakis, J. Universal Frequency-Dependent AC Conductivity of Conducting Polymer Networks. *Appl. Phys. Lett.* **91**, (2007).
5. Kirchartz, T., Agostinelli, T., Campoy-Quiles, M., Gong, W. & Nelson, J. Understanding the Thickness-Dependent Performance of Organic Bulk Heterojunction Solar Cells: The Influence of Mobility, Lifetime, and Space Charge. *J. Phys. Chem. Lett.* **3**, 3470–3475 (2012).
6. Hu, L., Dagleish, S., Matsushita, M. M., Yoshikawa, H. & Awaga, K. Storage of an Electric Field for Photocurrent Generation in Ferroelectric-Functionalized Organic Devices. *Nat. Commun.* **5**, 3279 (2014).
7. Zhao, C., Li, C. & Duan, L. A Competitive Hopping Model for Carrier Transport in Disordered Organic Semiconductors. *Phys. Chem. Chem. Phys.* **21**, 9905–9911 (2019).
8. Jiang, H., Sun, J.-X. & Yang, H.-C. Dispersion Relation and General Charge-Transport Model for Organic Semiconductors. *J. Electron. Mater.* **48**, 2955–2961 (2019).
9. Pal, A. J., Österbacka, R., Källman, K.-M. & Stubb, H. Transient Electroluminescence: Mobility and Response Time in Quinquethiophene Langmuir–Blodgett Films. *Appl. Phys. Lett.* **71**, 228–230 (1997).
10. Novikov, S. V. Charge Carrier Diffusion in Energy Landscape Created by Static Charges: Poole–Frenkel Model Revised. *Phys. Status Solidi B* **236**, 119–128 (2003).
11. Yao, J. *et al.* Cathode Engineering with Perylene-Diimide Interlayer Enabling Over 17% Efficiency Single-Junction Organic Solar Cells. *Nat. Commun.* **11**, 2726 (2020).
12. Enhancing the Electron Blocking Ability of N-Type MoO₃ by Doping with P-Type NiO_x for Efficient Nonfullerene Polymer Solar Cells. *Org. Electron.* **68**, 168–175 (2019).
13. Röhr, J. A., Moia, D., Haque, S. A., Kirchartz, T. & Nelson, J. Exploring the

Validity and Limitations of the Mott–Gurney Law for Charge-Carrier Mobility Determination of Semiconducting Thin-Films. *J. Phys.: Condens. Matter* **30**, 105901 (2018).

14. Chang, Y., Zhu, X., Lu, K. & Wei, Z. Progress and Prospects of Thick-Film Organic Solar Cells. *J. Mater. Chem. A* **9**, 3125–3150 (2021).

15. Chen, H.-Y. *et al.* Polymer Solar Cells with Enhanced Open-Circuit Voltage and Efficiency. *Nat. Photonics* **3**, 649–653 (2009).

16. Jiang, D. *et al.* Extracting Charge Carrier Mobility in Organic Solar Cells Through Space-Charge-Limited Current Measurements. *Mater. Sci. Eng. R Rep.* **157**, 100772 (2024).

17. Wen, Z. *et al.* Influence of Donor:acceptor Ratio on Charge Transfer Dynamics in Non-Fullerene Organic Bulk Heterojunctions. *Chin. Chem. Lett.* **32**, 529–534 (2021).

18. Tsai, Y.-S. *et al.* Efficiency Improvement of Organic Solar Cells by Slow Growth and Changing Spin-Coating Parameters for Active Layers. *Jpn. J. Appl. Phys.* **50**, 022301 (2011).

19. Sun, R. *et al.* High-Speed Sequential Deposition of Photoactive Layers for Organic Solar Cell Manufacturing. *Nat. Energy* **7**, 1087–1099 (2022).

20. Yang, W. *et al.* Simultaneous Enhanced Efficiency and Thermal Stability in Organic Solar Cells from a Polymer Acceptor Additive. *Nat. Commun.* **11**, 1218 (2020).

21. Fan, Q. *et al.* High-Performance As-Cast Nonfullerene Polymer Solar Cells with Thicker Active Layer and Large Area Exceeding 11% Power Conversion Efficiency. *Adv. Mater.* **30**, 1704546 (2018).

22. Yuan, J. *et al.* Single-Junction Organic Solar Cell with over 15% Efficiency Using Fused-Ring Acceptor with Electron-Deficient Core. *Joule* **3**, 1140–1151 (2019).

23. Liu, Z. *et al.* 15.28% Efficiency of Conventional Layer-by-Layer All-Polymer Solar Cells Superior to Bulk Heterojunction or Inverted Cells. *Chem. Eng. J.* **450**, 138146 (2022).

24. Chen, C. *et al.* Molecular Interaction Induced Dual Fibrils Towards Organic Solar Cells with Certified Efficiency Over 20%. *Nat. Commun.* **15**, 6865 (2024).

25. Zhan, L. *et al.* Multiphase Morphology with Enhanced Carrier Lifetime via

Quaternary Strategy Enables High-Efficiency, Thick-Film, and Large-Area Organic Photovoltaics. *Adv. Mater.* **34**, 2206269 (2022).

Reviewer #2 (Remarks to the Author):

Meng and coauthors demonstrate a new model for screening photovoltaic materials for thick-film Organic Solar Cells (OSCs), introducing "critical length" as a key screening metric. This approach addresses a long-standing limitation of relying solely on carrier mobility. The authors claim that the critical length is a key factor for a range of acceptor materials, including both fullerene and non-fullerene materials, and identify BTP-eC9 and L8-BO as promising candidates. Subsequently, the authors fabricate a series of OSCs based on BTP-eC9, achieving a high Power Conversion Efficiency (PCE) exceeding 19% in thick-film OSCs. The concept of critical length provides a novel perspective that may help guide the design of thick-film devices. The critical length introduced in this article incorporates various steady-state properties, including mobility and AC conductivity. Performance correlations for thick-film devices are effectively established through the testing of these parameters. This novel experimental approach, linking microstructural morphology to overall device performance through classical characterization techniques, holds significant importance. However, the current presentation lacks details for this central part of the work. It is recommended that the authors expand and clearly discuss this section, enabling a broader audience of researchers in the field to become familiar with the method. This will enhance both the accessibility and the appeal of the article to a wider readership. Overall, this work is nicely presented, and thus, this work could be published in *Nature Communications* following the revisions commented below:

- (1) *Figure 1c represents a correlation between the critical length and the average carrier velocity, but there is no term related to the average velocity in Equation 1, so Equation 1 needs further explanation.*

Response: We appreciate the reviewer's valuable comments. In **Figure 1c**, we present a 3D plot to more intuitively illustrate the relationship among critical length, average velocity, and hopping frequency. This plot is based on the original definition of critical length from the literature, represented as a three-dimensional function. We assigned

reasonable values to the average velocity and hopping frequency parameters so that the resulting values correspond closely to those measured in our organic systems. The final schematic is shown in **Figure 1c**.

The concept of critical length was originally proposed by Papathanassiou et al. to describe the distribution of conductive pathways in the universal AC conductivity response model. It was defined as the ratio of the average carrier velocity to the hopping frequency (or equivalently, the inverse of the hopping time).¹

Charge carrier transport in the active layer is governed, on one hand, by the intrinsic transport - namely the zero-field mobility (μ_0), which reflects the fundamental capability of carriers to move under vanishing fields.² On the other hand, the overall energy landscape fluctuations induced by defect states and static charges - through Coulombic potential fluctuations - play a crucial role in determining the carrier's migration and diffusion pathways. Field-dependent of mobility β reflects the degree to which the external electric field lowers the energy barrier for a carrier to escape from an isolated trap and is closely related to the dielectric properties of the material, as shown in the Poole-Frenkel formalism. A smaller β implies a weaker field dependence, indicating that the mobility remains relatively high even under low-field conditions, which is favorable for long-range charge carrier transport. In contrast, a larger β signifies that the carrier mobility drops significantly in weak fields, hindering long-range transport. From a broader perspective, the value of β serves as a proxy for the smoothness or ruggedness of the energy landscape. A large β reflects more pronounced energetic disorder and stronger field dependence of mobility. This interpretation is consistent with the revised PF framework where β characterizes how the underlying energy disorder governs field-activated transport.³ Moreover, recent studies have demonstrated that field-dependent of mobility significantly affects key performance parameters such as the fill factor (FF) and power conversion efficiency (PCE), especially in thick-film devices.

Therefore, we combined the intrinsic mobility (μ_0) and field-dependent of mobility (β) to construct a meaningful characteristic velocity, which is used to estimate the spatial transport scale of carriers under a Poole–Frenkel hopping mechanism. This constructed velocity can capture the transport dynamics in OSCs, where the charge carriers are subject to non-steady electric fields and a highly disordered energy landscape.

Additionally, in AC conductivity measurements, the applied voltage is sinusoidal, alternating between forward and reverse field directions. These opposing field components respectively promote and hinder carrier extraction at the cathode. To account for this in the critical length estimation, we consider only the forward half-cycle of the sinusoidal field, and therefore adopt half the period ($t=1/2f=1/2\omega$) in the denominator. As a result, the full expression for the critical length includes the factor $2\omega_H$ in the denominator.

In summary, the critical length of charge carriers in organic solar cells is calculated using Equation 1 in the manuscript. On page 6 of the revised manuscript, we have added the following relevant discussion, and added the detailed derivation of Equation (1) on pages 8 and 9 of the revised ESI.

(2) From Equation 1, the authors suggest a low hopping frequency is needed to obtain a large critical length, which contradicts the traditional expectation that a low hopping frequency means lower mobility. Clarifying this point will help readers better understand the physical implications of the model.

Response: We appreciate the reviewer for the valuable comment. The measurement of AC conductivity involves applying a high-frequency electric field that restricts carrier motion to hopping between localized states. Using the Almond-West formula, the carrier hopping frequency can thus be determined. However, our analysis of the correlation between hopping frequency and mobility, as shown in **Figure 2e** of the manuscript, reveals no clear positive correlation between these two parameters. This is

primarily because hopping frequency reflects the localized hopping of charge carriers among nearby states, whereas mobility encompasses the overall transport across the entire film. Mobility is influenced by multiple factors including transport pathways, trap states, and carrier recombination. Unlike hopping frequency, mobility is a more macroscopic parameter requiring carriers to traverse the full film to contribute to measurable current. The critical length is defined as “the maximum distance a charge carrier can travel at an average velocity within a single hopping time.” The reciprocal of the hopping frequency corresponds to the charge carrier hopping time, during which recombination or trapping is assumed negligible. Consequently, a longer hopping time (i.e., lower hopping frequency) results in a larger critical length. We have added the original definition of the critical length on pages 8 and 9 of the revised ESI, along with the above expanded explanation to clarify the basis of our approach.

(3) The wavelength ranges for extracting localized excitons, hole transfer, and other transient spectral signals must be clearly specified in transient spectral characterization.

Response: We appreciate the reviewer for the valuable comment. In transient absorption spectroscopy characterization, we detected localized exciton signals at 860 nm and 912 nm for the D18:L8-BO:BTP-eC9 and D18:L8-BO:N3 systems, respectively. Delocalized exciton signals were observed at 1432 nm for the D18:N3:BTP-eC9 system. The average exciton decay times were extracted by fitting the decay curves with exponential functions.

In the visible region, we extracted the donor ground-state bleach signal at 590 nm to analyze the hole transfer rate. The wavelengths used for detection are marked on page 16 of the revised manuscript.

Figure 4. Femtosecond resolved transient absorption spectroscopy (TAS) results. (a) 2D TA data of D18:L8-BO. (b) 2D TA data of D18:L8-BO:8.3% BTP-eC9. (c) Localized exciton decay signals extracted from 860 nm. (d) Variation of average decay time of localized exciton with BTP-eC9 content. (e) TA spectra at different delay times of D18:L8-BO. (f) TA spectra at different delay times of D18:L8-BO:8.3% BTP-eC9. (g) Kinetic traces at the selected wavelength of D18:L8-BO (The ground-state bleaching (GSB) signal of the donor is located at 590 nm). (h) Kinetic traces at the selected wavelength of D18:L8-BO:8.3% BTP-eC9.

(4) *It is necessary to clarify which recombination mechanism dominates in the D18:N3 and D18:L8-BO systems, respectively, regarding carrier recombination mechanisms.*

Response: We appreciate the reviewer for the valuable comment. We characterized charge carrier dynamics in two high-efficiency thick-film device systems, D18:N3 and D18:L8-BO. Through light-intensity-dependent short-circuit current density measurements, we investigated bimolecular recombination. In the D18:N3 system, the slope remained at 0.99 before and after adding BTP-eC9 (Table S12 in the ESI), indicating that bimolecular recombination is not dominant.

Using light-intensity-dependent open-circuit voltage measurements, we explored trap-assisted recombination. The slope n decreased from 1.87 to 1.64 upon adding BTP-eC9

in the D18:N3 system (**Table S13** in the ESI), suggesting suppression of trap-assisted recombination. In contrast, for the D18:L8-BO system, both the slopes of light-intensity-dependent short-circuit current density and open-circuit voltage decreased after BTP-eC9 addition, indicating that both bimolecular and trap-assisted recombination play significant roles in this system. We have added the relevant discussion on page 17 of the revised manuscript.

(5) *Several inconsistencies are noted in the formatting of the references, particularly in the use of capitalization and subscripts/superscripts in titles (e.g., Reference 22 in the supplementary file). It is recommended that the references be carefully reviewed and formatted in accordance with the journal's guidelines prior to resubmission.*

Response: We appreciate the reviewer for the valuable comment. We have corrected the errors found in the references as pointed out.

References of manuscript:

“8. Dong, Q. *et al.* Electron-Hole Diffusion Lengths > 175 μm in Solution-Grown $\text{CH}_3\text{NH}_3\text{PbI}_3$ Single Crystals. *Science* **347**, 967–970 (2015).

31. Dhiman, S., Meena, R., Manyani, N. & Tripathi, S. K. Investigating the Temperature and Frequency Dependence of Dielectric Response Using AC Impedance Spectroscopy on SnO_2 . *Surf. Interfaces* **42**, 103362 (2023).”

Supplementary References:

“18. Fenta, A. D., Lu, C.-F., Gidey, A. T. & Chen, C.-T. High Efficiency Organic Photovoltaics with a Thick (300 nm) Bulk Heterojunction Comprising a Ternary Composition of a PFT Polymer– PC_{71}BM Fullerene–IT4F Nonfullerene Acceptor. *ACS Appl. Energy Mater.* **4**, 5274–5285 (2021).

23. Yoon, S. *et al.* High-Performance Scalable Organic Photovoltaics with High Thickness Tolerance from 1 cm^2 to Above 50 cm^2 . *Joule* **6**, 2406–2422 (2022).

24. Liu, Z. & Wang, H.-E. High-Performance Ternary Organic Photovoltaics with NC₇₀BA as the Third Component Material Enabling Thickness-Insensitive Photoactive Performance. *Nanotechnology* **33**, 065206 (2021).
32. Dai, T. *et al.* Reduced Exciton Binding Energy and Diverse Molecular Stacking Enable High-Performance Organic Solar Cells with V_{OC} Over 1.1 V. *Sci. China Chem.* **67**, 3140–3152 (2024).
59. Wang, H., Wang, X., Fan, P., Yang, X. & Yu, J. Enhanced Power Conversion Efficiency of P3HT : PC₇₁BM Bulk Heterojunction Polymer Solar Cells by Doping a High-Mobility Small Organic Molecule. *Int. J. Photoenergy* **2015**, 982064 (2015).
88. Liao, C. *et al.* Tetrahydrofuran Processable Organic Solar Cells with 19.45% Efficiency Realized by Introducing High Molecular Dipole Unit into the Terpolymer. *Adv. Mater.* **36**, 2411071 (2024).”

(6) *The improvement in device performance is attributed to increases in short-circuit current density (J_{sc}) and fill factor (FF). What are the primary factors responsible for the improvement in FF? Please provide a systematic explanation.*

Response: We appreciate the reviewer for the valuable comment. We introduced BTP-eC9, an acceptor with a large critical length, as a third component into different systems to improve the performance of thick-film devices. This enhancement is mainly attributed to increases in short-circuit current density and fill factor.

We performed defect-state, carrier recombination, and energetic disorder characterizations on the highest-performing devices: D18:N3:BTP-eC9 and D18:L8-BO:BTP-eC9 systems. From the defect-state analysis, the addition of BTP-eC9 reduced the defect density in both systems (see **Table S11** in the ESI), which decreases the probability of carrier trapping. Regarding carrier recombination, trap-assisted recombination was suppressed in the D18:N3 system after adding BTP-eC9, while both bimolecular and trap-assisted recombination were reduced in the D18:L8-BO system. The overall reduction in recombination enhances the probability of carriers being

collected by the electrodes.

Furthermore, the incorporation of BTP-eC9 suppressed the tail-state density within the bandgap, as shown in **Table S16** of the ESI. The reductions in defect density, carrier recombination, and tail-state density are reflected in changes to carrier extraction time and carrier lifetime. Specifically, carrier extraction becomes faster and carrier lifetime is extended (see **Tables S14** and **S15** in the ESI).

(7) The morphological features of the blend systems have been characterized. It is essential to clarify which morphological features contribute to the improvement of J_{sc} and FF mentioned in the article.

Response: We appreciate the reviewer for the valuable comment. We introduced BTP-eC9, an acceptor with a large critical length, as a third component into different systems to enhance the performance of thick-film devices. This improvement is primarily attributed to increased short-circuit current density and fill factor.

We performed comprehensive morphology analyses of the active layers with added BTP-eC9 using depth-dependent absorption spectroscopy, GISAXS, and AFM, as shown in **Figures 6** and **7** of the manuscript. From the vertical donor–acceptor phase distribution, the addition of BTP-eC9 reduced donor content near the cathode interface while increasing acceptor concentration in this region, which helps suppress carrier recombination at the interface. GISAXS analysis revealed that BTP-eC9 incorporation enlarged the acceptor domain size, creating favorable conditions for long-range charge carrier transport.

These findings are consistent with previous studies reporting that increased acceptor domain size and extended charge carrier transport distances contribute to higher fill factors.^{4,5} We have provided a relevant discussion on page 22 of the revised manuscript.

(8) A few minor corrections are needed. For example, when referring to multiple figures, plural forms should be used (e.g., "Figures 1 and 2"). In addition, the x-axis labels in Figure 7 appear too close to the axis and may require better spacing for readability.

Response: We appreciate the reviewer for the valuable comment. We have corrected grammatical errors in the manuscript and revised the X-axis labeling in Figure 7 accordingly.

Figure 7. Acceptor domain size analysis. (a) GISAXS image of PM6:N3. (b) GISAXS image of PM6:N3:BTP-eC9. (c) GISAXS image of PM6:N4. (d) GISAXS image of PM6:N4:BTP-eC9. (e) GISAXS image of PM6:BPF-4F. (f) GISAXS image of PM6:BPF-4F:BTP-eC9. (g) GISAXS image of D18:N3. (h) GISAXS image of D18:N3:BTP-eC9. (i) GISAXS image of D18:N4. (j) GISAXS image of D18:N4:BTP-eC9. (k) GISAXS image of D18:L8-BO. (l) GISAXS image of D18:L8-BO:BTP-eC9. (m) Changes in the acceptor domain size after adding BTP-eC9 to active layer with a

thickness of 100 nm. (n) Changes in the acceptor domain size after adding BTP-eC9 to active layer with a thickness of 300 nm.

References:

1. Papathanassiou, A. N., Sakellis, I. & Grammatikakis, J. Universal Frequency-Dependent AC Conductivity of Conducting Polymer Networks. *Appl. Phys. Lett.* **91**, (2007).
2. Pal, A. J., Österbacka, R., Källman, K.-M. & Stubb, H. Transient Electroluminescence: Mobility and Response Time in Quinquethiophene Langmuir–Blodgett Films. *Appl. Phys. Lett.* **71**, 228–230 (1997).
3. Novikov, S. V. Charge Carrier Diffusion in Energy Landscape Created by Static Charges: Poole–Frenkel Model Revised. *Phys. Status Solidi B* **236**, 119–128 (2003).
4. Hamilton, R. *et al.* Recombination in Annealed and Nonannealed Polythiophene/Fullerene Solar Cells: Transient Photovoltage Studies versus Numerical Modeling. *J. Phys. Chem. Lett.* **1**, 1432–1436 (2010).
5. Yan, J. *et al.* Optimized Phase Separation and Reduced Geminate Recombination in High Fill Factor Small-Molecule Organic Solar Cells. *ACS Energy Lett.* **2**, 14–21 (2017).

Reviewer #3 (Remarks to the Author):

In this study, the authors propose the critical length as a predictor for thick-film device performance and demonstrate its broad applicability through a detailed investigation of various acceptor materials. This method represents a meaningful advancement toward the scalable fabrication of high-performance organic photovoltaics (OPVs). Importantly, the study establishes a quantitative relationship between critical length and film morphology (particularly with domain size), offering a practical and visual guideline for optimizing thick-film device performance. Using such a method, the authors achieve a record-high power conversion efficiency (PCE) in thick-film devices. The charge transport modeling with device fabrication and morphological results in a coherent and compelling data description. Therefore, I am pleased to recommend an acceptance decision for this manuscript to be published in *Nature Communications* after minor revisions. Authors need to fully address the required issues listed below:

1. *The manuscript attempts to introduce the physical basis of the critical length concept; however, it remains to be clarified whether this concept applies to more complex systems involving organic donor-acceptor blends. Clarification is needed on whether the model addresses static (energetic) or dynamic (vibrational) disorder.*

Response: We appreciate the reviewer for the valuable comment. The study of AC conductivity spans both ionic and electronic conduction, with a common characteristic of these solids being structural disorder. Examples include disordered semiconductors, polymers, conductive polymer composite ceramics, ion-conducting glasses, and heavily doped ionic crystals. The universal response of AC conductivity refers to the frequency-dependent conductivity spectrum, where the low-frequency region exhibits a constant conductivity value known as the DC conductivity. In the high-frequency region, the conductivity shows an exponential dependence on frequency due to the back-and-forth motion of charge carriers, termed AC conductivity.

In organic solar cells (OSCs), charge carrier transport performance is significantly

influenced by material disorder, which can be classified into static disorder (structural heterogeneity) and dynamic disorder (time-dependent structural fluctuations). Static disorder arises from inherent heterogeneity within the material, such as molecular packing disorder, defect distribution, and phase-separation morphology, and generally remains constant over time. This static disorder leads to a broadened density of states (DOS), requiring carriers to hop between localized states of varying energies, which increases the activation energy for hopping transport and reduces mobility.¹ Disordered regions may form deep or shallow trap states, where trapped carriers recombine non-radiatively, thereby reducing charge collection efficiency.

Dynamic disorder results from molecular vibrations, conformational changes, or environmental polarization effects and is time-dependent. Appropriate levels of dynamic disorder can assist carriers in escaping localized states through thermal fluctuations.²

The universal response of AC conductivity emerges from the combined effects of static and dynamic disorder; however, their dominance varies across different frequency ranges and temperatures. In the low-frequency region, carrier hopping conduction is primarily governed by static disorder due to localized states, while energy fluctuations caused by dynamic disorder may assist carriers in overcoming static disorder barriers, influencing the high-frequency response through energy fluctuations and polarization effects. The synergy of static and dynamic disorder ultimately determines the overall charge transport characteristics: static disorder limits long-range mobility, whereas dynamic disorder modulates short-range hopping. We have added the relevant discussion on page 8 of the revised ESI.

- 2. The discussion on the critical length is informative. However, what is the definition or extraction method used to determine the hopping frequency? Additionally, are there alternative models that could be considered for determining the hopping frequency?*

Response: In this work, we employed AC conductivity measurements to extract the carrier hopping frequency. At low frequencies, carriers undergo long-range transport across the entire sample and are collected by the electrodes, resulting in a constant DC conductivity. As the electric field frequency increases, the effective charge carrier transport path length becomes limited, and in the high-frequency region, carriers predominantly perform hopping between localized states. Consequently, conductivity exhibits an exponential increase with frequency.

Almond derived the Almond-West formula by considering the effect of carrier hopping on conductivity, which is widely used to fit AC conductivity data. In disordered solids, the DC conductivity is determined by the carrier concentration and hopping frequency, all of which exhibit thermally activated behavior. Almond found that the activation energy for DC conductivity matches that of the hopping frequency, indicating that both DC and AC conductivity arise from the same carrier hopping mechanism. Notably, carrier concentration does not affect the formation of AC conductivity.

Therefore, the Almond-West formula enables extraction of the carrier hopping frequency from the transition point between DC and AC conductivity in disordered solids. We have added the relevant discussion on page 6 of the revised manuscript.

3. *The article presents extensive characterization data for both the D18:L8-BO and D18:N3 systems. What is the common impact of high critical length acceptor materials on device performance in both systems?*

Response: We appreciate the reviewer for the valuable comment. In this work, we proposed a critical length screening framework to identify suitable acceptor materials for thick-film devices. Previously, through this screening, we identified BTP-eC9 as an acceptor with a large critical length and high potential for fabricating efficient thick-

film organic solar cells. Therefore, we incorporated BTP-eC9 as a third acceptor component into various systems to fabricate high-performance thick-film devices.

Analysis of device data based on different donor systems, including D18 and PM6, shows that adding BTP-eC9 improved the power conversion efficiency (PCE) by approximately 3%, mainly due to enhancements in fill factor and short-circuit current density. We further characterized defect-state density, carrier recombination, and tail-state density within the bandgap. After incorporating BTP-eC9, defect-state density was suppressed, trap-assisted recombination was reduced in the D18:N3 system, while both bimolecular and trap-assisted recombination were suppressed in the D18:L8-BO system. Tail-state density decreased in both systems. GISAXS and AFM characterizations revealed that adding BTP-eC9 promoted the formation of longer, continuous aggregated domains in the active layer (see **Figure 6** in the manuscript), resulting in enlarged acceptor domain sizes.

Meanwhile, the reduction in defect density, suppression of carrier recombination, and extension of acceptor domain size collectively create favorable conditions for long-range charge carrier transport, which in turn contributes to the improvements in short-circuit current density and fill factor observed in the devices.

4. *The critical length presented in the manuscript is determined by three parameters, each reflecting certain characteristics of charge carriers. To enhance the physical insight and clarity of the model, it would be beneficial to provide more specific physical interpretations of these parameters in the context of charge transport mechanisms.*

Response: We appreciate the reviewer for the valuable comment. In our manuscript, we proposed a critical length screening framework to identify suitable acceptor materials for thick-film devices. The concept of critical length originates from Papathanassiou et al.'s work on the universal response model of AC conductivity, where

it is defined based on the distribution of conductive pathways. Its original definition is given by the formula below, in which the critical length equals the average velocity divided by the hopping time.

In OSCs, due to the difference in work functions between cathode and anode materials, a built-in electric field is established inside the device. The disordered states within the active layer cause the charge carrier transport to exhibit field-dependent behavior. This built-in field causes the actual charge carrier transport to deviate from that described by zero-field mobility, and zero-field mobility fitting can vary significantly depending on the model used.

Based on these considerations, we have refined the Papathanassiou model's critical length to better account for the deviation of charge carrier transport in thick-film devices from the zero-field mobility description. By combining the mobility and its field dependence, we introduce a characteristic average velocity. The resulting critical length thus integrates the hopping frequency (reflecting carrier hops between localized states), the mobility (representing macroscopic carrier drift speed), and the field dependence of mobility.

Therefore, this modified critical length provides a more comprehensive metric for evaluating charge carrier transport in thick-film devices, as it simultaneously describes carrier hopping at the microscopic level, long-range transport velocity, and the impact of the device's built-in field, offering a more complete picture than using zero-field mobility alone. We have added the relevant discussion on page 8 of the revised ESI.

5. *Transient absorption spectroscopy analyzes exciton and hole transfer dynamics and identifies the optimal BTP-eC9 ratio in both systems. Is this ratio consistent with the actual ratio used during device fabrication? The manuscript should clearly state the specific amount of BTP-eC9 added in the device preparation process.*

Response: We appreciate the reviewer for the valuable comment. We used transient absorption spectroscopy to investigate the effects of varying BTP-eC9 ratios on exciton dynamics and hole transfer kinetics in the D18:N3 and D18:L8-BO systems.

In the visible region, we analyzed hole transfer rates in the active layer via the donor ground-state bleach signal. For the D18:L8-BO system, the fastest hole transfer rate was observed with 8.3% BTP-eC9 addition.

In the near-infrared region, analysis of the exciton signals revealed that the fastest exciton decay occurred with 30% BTP-eC9 in the D18:N3 system, and with 8.3% BTP-eC9 in the D18:L8-BO system, indicating the most efficient exciton dissociation. This trend aligns well with the BTP-eC9 ratios that yielded the highest device efficiencies in our fabrication process.

We have included the specific BTP-eC9 addition ratios on page 3 of revised ESI: “In the D18:L8-BO:BTP-eC9 system, the total donor-to-acceptor ratio is 1:1.2, with the acceptor components L8-BO and BTP-eC9 mixed at a ratio of 1.1:1. In the D18:N3:BTP-eC9 system, the overall donor-to-acceptor ratio is also 1:1.6, where the acceptor blend ratio of N3 to BTP-eC9 is 1.12:0.48.”

References:

1. Zhang, C. *et al.* Unraveling Urbach Tail Effects in High-Performance Organic Photovoltaics: Dynamic vs Static Disorder. *ACS Energy Lett.* **7**, 1971-1979 (2022).
2. Peng, X., Li, Q. & Shuai, Z. Influences of Dynamic and Static Disorder on the Carrier Mobility of BTBT-C12 Derivatives: A Multiscale Computational Study. *Nanoscale* **13**, 3252–3262 (2021).